# Clinical interventions for adults with comorbid alcohol use and depressive disorders: A systematic review and network meta-analysis

**Sean Grant** [1,2]*, **Gulrez Azhar**[2], **Eugeniu Han**[2], **Marika Booth**[2], **Aneesa Motala**[2], **Jody Larkin**[3], **Susanne Hempel**[2]

**1** Department of Social & Behavioral Sciences, Indiana University Richard M. Fairbanks School of Public Health, Indianapolis, Indiana, United States of America, **2** RAND Corporation, Santa Monica, California, United States of America, **3** RAND Corporation, Pittsburgh, Pennsylvania, United States of America

* spgrant@iu.edu

## Abstract

### Background

Uncertainty remains regarding the effectiveness of treatments for patients diagnosed with both an alcohol use disorder (AUD) and depressive disorder. This study aimed to compare the effectiveness of clinical interventions for improving symptoms of adults with co-occurring AUDs and depressive disorders.

### Methods and findings

We searched CINAHL, ClinicalTrials.gov, Cochrane Central Register of Controlled Trials, Cochrane Database of Systematic Reviews, Excerpta Medica Database, International Clinical Trials Registry Platform (ICTRP), PubMed, PsycINFO, and Web of Science from inception to December 2020. We included randomized controlled trials (RCTs) evaluating clinical interventions for adults with co-occurring AUDs and depressive disorders. Two independent reviewers extracted study-level information and outcome data. We assessed risk of bias using the Cochrane Risk of Bias tool, used frequentist random effects models for network meta-analyses, and rated our confidence in effect estimates using the Grading of Recommendations Assessment, Development, and Evaluation (GRADE) approach. Primary outcomes were remission from depression and alcohol use. Secondary outcomes were depressive symptoms, alcohol use, heavy drinking, health-related quality of life, functional status, and adverse events. We used standardized mean differences (SMDs) for continuous outcomes and odds ratios (ORs) for dichotomous outcomes to estimate intervention effects. Overall, 36 RCTs with 2,729 participants evaluated 14 pharmacological and 4 psychological interventions adjunctive to treatment as usual (TAU). Studies were published from 1971 to 2019, conducted in 13 countries, and had a median sample size of 50 participants (range: 14 to 350 participants). We have very low confidence in all estimates of intervention effects on our primary outcomes (i.e., remission from depression and remission from alcohol use). We have moderate confidence that cognitive behavioral therapies (CBTs) demonstrated greater benefit than no additional treatment (SMD = −0.84; 95% confidence interval [CI],

**Data Availability Statement:** All relevant data are within the manuscript and its Supporting information files.

**Funding:** This review was funded by the US Department of Defense Psychological Health Center of Excellence (PI: SH). The funders provided feedback during study design on the research questions of the review and data to be collected. The funders had no role in data collection and analysis, decision to publish, or preparation of the manuscript.

**Competing interests:** We have read the journal's policy and the authors of this manuscript have the following competing interests: SG's spouse is a salaried-employee of Eli Lilly and Company, and owns stock. SG has accompanied his spouse on company-sponsored travel. All other authors have declared that no competing interests exist.

**Abbreviations:** AUD, alcohol use disorder; CBT, cognitive behavioral therapy; CI, confidence interval; DSM, Diagnostic and Statistical Manual of Mental Disorders; GRADE, Grading of Recommendations Assessment, Development, and Evaluation; ICD, International Classification of Diseases; ICTRP, International Clinical Trials Registry Platform; NMA, network meta-analysis; OR, odds ratio; PRISMA-P, Preferred Reporting Items for Systematic Reviews and Meta-Analyses Protocols; RCT, randomized controlled trial; SMD, standardized mean difference; SSRI, selective serotonin reuptake inhibitor; TAU, treatment as usual; TCA, tricyclic antidepressant.

$-1.05$ to $-0.63$; $p < 0.001$) for depressive symptoms and low confidence (SMD = $-0.25$; 95% CI, $-0.47$ to $-0.04$; $p = 0.021$) for alcohol use. We have low confidence that tricyclic antidepressants (TCAs) demonstrated greater benefit than placebo (SMD = $-0.37$; 95% CI, $-0.72$ to $-0.02$, $p = 0.038$) for depressive symptoms. Compared with placebo, we have moderate confidence that selective serotonin reuptake inhibitors (SSRIs) demonstrated greater benefit for functional status (SMD = $-0.92$; 95% CI, $-1.36$ to $-0.47$, $p < 0.001$) and low confidence for alcohol use (SMD = $-0.30$; 95% CI, $-0.59$ to $-0.02$, $p = 0.039$). However, we have moderate confidence that patients receiving SSRIs also were more likely to experience an adverse event (OR = 2.20; 95% CI, 0.94 to 5.16, $p = 0.07$). We have very low confidence in all other effect estimates, and we did not have high confidence in any effect estimates. Limitations include the sparsity of evidence on intervention effects over the long term, risks of attrition bias, and heterogeneous definitions of adverse events in the evidence base.

## Conclusions

We are very uncertain about the existence (or not) of any non-null effects for our primary outcomes of remission from depression and remission from alcohol use. The available evidence does suggest that CBTs likely reduced, and TCAs may have resulted in a slight reduction of depressive symptoms. SSRIs likely increased functional status, and SSRIs and CBTs may have resulted in a slight reduction of alcohol use. However, patients receiving SSRIs also likely had an increased risk of experiencing an adverse event. In addition, these conclusions only apply to postintervention and are not against active comparators, limiting the understanding of the efficacy of interventions in the long term as well as the comparative effectiveness of active treatments. As we did not have high confidence in any outcomes, additional studies are warranted to provide more conclusive evidence.

## Author summary

### Why was this study done?

- Alcohol use disorders (AUDs) and depressive disorders are prevalent behavioral health conditions among adult populations, often co-occur, and have significant personal, societal, and economic consequences.

- Existing systematic reviews and clinical practice guidelines often focus on either AUDs or depressive disorders, despite the prevalence and significance of their co-occurrence.

- The objective of this review is to examine the available evidence on the effectiveness of clinical interventions for adult patients with co-occurring AUD and depressive disorders.

### What did the researchers do and find?

- We conducted a systematic review and network meta-analysis (NMA) of 36 randomized controlled trials (RCTs) with 2,729 participants evaluating 14 pharmacological and 4

psychological interventions for adults with co-occurring AUDs and depressive disorders.

- We have very low confidence in all estimates of intervention effects on our primary outcomes (i.e., remission from depression and remission from alcohol use).

- We found that cognitive behavioral therapies (CBTs) likely reduced, and tricyclic antidepressants (TCAs) may have resulted in a slight reduction of depressive symptoms, selective serotonin reuptake inhibitors (SSRIs) likely increased functional status, SSRIs and CBTs may have resulted in a slight reduction of alcohol use, and SSRIs also likely resulted in an increased risk of experiencing an adverse event.

- We have very low confidence in all other effect estimates, and we did not have high confidence in any effect estimates.

### What do these findings mean?

- We did not have high confidence in any effect estimates, and we have very low confidence in the vast majority of estimates of intervention effects across all outcomes.

- For policy and practice, we are very uncertain about the existence (or not) of any non-null effects for our primary outcomes of remission from depression and remission from alcohol use. The available evidence does suggest potentially actionable benefits at postintervention of CBTs for depressive symptoms and alcohol use, TCAs for depressive symptoms, and SSRIs for alcohol use and functional status—although SSRIs also likely have higher risks of adverse events (including serious adverse events).

- For research, future trials are needed that are prospectively registered, adequately powered, fit for pragmatic purposes, comprehensively report study information and outcomes, and evaluate interventions discussed in clinical practice guidelines yet missing from the current body of evidence.

## Introduction

Alcohol use disorders (AUDs) and depressive disorders are prevalent behavioral health conditions among adult populations with significant personal, societal, and economic consequences. Best estimates of current rates (past 12 months) for noninstitutionalized populations indicate that 13.9% of adults meet criteria for an AUD, and 6.7% of adults meet criteria for a major depressive episode [1,2]. Adults with an AUD are more likely than those without an AUD to have worse physical health, mental health, and social functioning [2], while depression is one of the leading causes of disease burden worldwide and is associated with significantly increased risks of morbidity and mortality [3–5].

AUDs and depressive disorders often co-occur. Adults with any AUD (mild, moderate, or severe) in the past 12 months have 1.2 (95% confidence interval [CI] 1.08 to 1.35) times the odds of having a major depressive disorder compared with adults without an AUD [2]. Co-occurring AUD and depression results in worse treatment outcomes on average compared with patients diagnosed with only one of these disorders [6]. However, current clinical practice

guidelines often focus on one or the other type of disorder, despite the prevalence and significance of their co-occurrence [7,8]. Previous systematic reviews provide empirical support for numerous psychological and pharmacological interventions for the treatment of patients with either an AUD [9,10] or a depressive disorder [11–13]. Rigorous evidence is needed regarding the use of these interventions to treat patients with both an AUD and a depressive disorder [6,14]. The objective of this review is to examine the available evidence on the effectiveness of clinical interventions for adult patients with co-occurring AUD and depressive disorders. To achieve this objective, we performed a network meta-analysis (NMA). An NMA combines both direct and indirect comparisons of intervention effects, obtaining an effect estimate for each possible pair of interventions (including those that have not been directly compared). Consequently, identifying and synthesizing evidence from the entire network of evidence enable a more comprehensive understanding of the comparative effectiveness of interventions for a given population and outcome. As such, it is a powerful research tool to assist patients, providers, and policymakers to make informed decisions about which intervention is most likely to improve healthcare at the individual and population levels.

## Methods

We registered the protocol for this review in the international prospective register of systematic reviews before completing formal screening of search results against eligibility criteria (PROSPERO identifier CRD42017078239). We prepared the protocol and this report using the relevant Preferred Reporting Items for Systematic Reviews and Meta-Analyses Protocols (PRISMA-P) 2015 Statement [15,16], as well as the Methodological Expectations of Cochrane Intervention Reviews [17]. This study is reported according to the PRISMA Extension Statement for systematic reviews incorporating network meta-analyses (see S1 PRISMA Checklist) [18]. Further information regarding the methods and materials is available in the Supporting information (see S1 Text).

### Identification and selection of studies

We searched CINAHL, ClinicalTrials.gov, Cochrane Central Register of Controlled Trials, Cochrane Database of Systematic Reviews, Excerpta Medica Database, International Clinical Trials Registry Platform (ICTRP), PubMed, PsycINFO, and Web of Science for English language articles from inception to December 2020. A reference librarian for RAND's Knowledge Services (JL) developed the search strings (using search terms related to alcohol use, depression, and randomized trials) in consultation with the lead and senior authors (SG and SH) using terms identified in previous reviews on interventions for AUDs and depressive disorders [6,14,19–23]. We also reference mined the bibliographies of previous systematic reviews. Two reviewers (SG and either GA or EH) independently screened all titles and abstracts of retrieved citations. We conducted full-text eligibility assessment for citations judged as potentially eligible by at least 1 reviewer; we resolved any disagreements between the 2 reviewers about full-text eligibility through discussion within the review team.

We included parallel group (individually or cluster) randomized controlled trials (RCTs) only. Studies had to include adult participants (at least 50% were 18 years of age or older) with clinical diagnoses for both an AUD and depressive disorder according to Diagnostic and Statistical Manual of Mental Disorders (DSM) or International Classification of Diseases (ICD) criteria. In addition to formal diagnostic procedures, we also included studies that used non-operationalized diagnostic criteria, validated clinician-reported symptom questionnaires, or self-reported symptom questionnaires with established thresholds to identify patients with eligible diagnoses. For research conducted prior to DSM-III (i.e., before 1980), we included

studies in which investigators, study clinicians, and/or rating scales designated patients as having both "depression" and "alcoholism." Clinical interventions from any therapeutic approach were eligible so long as the evaluated intervention was intended to improve depressive symptoms or reduce alcohol use. Primary outcomes were remission from depression and alcohol use. Secondary outcomes were depressive symptoms, alcohol use, heavy drinking, health-related quality of life, functional status, and adverse events. We did not exclude studies based on comparator interventions, follow-up period for outcome assessment, setting, publication status, or publication language.

A crucial aspect of NMAs involves visualizing the interventions that have been evaluated for a population of interest as forming a network in which the interventions are represented by dots (or "nodes") and comparisons between interventions are represented by lines (or "edges") in a diagram. After completing the search but before extracting and analyzing outcome data, we assigned identified interventions to nodes in our network via consensus among the review team and external advisers, using a preregistered list of intervention nodes (see S1 Text) as a guide [24].

## Data extraction

We collected participant data based on the PRISMA-Equity Extension [25,26], intervention and comparator data using the template for intervention description and replication [27], outcome data using ClinicalTrials.gov criteria for completed defined outcomes [28], study setting data using the Consolidated Framework for Implementation Research [29], and study design data using the revised Cochrane tool [30]. Two reviewers independently extracted study-level descriptive data (SG and either GA or EH) and outcome data (SG and MB). We assessed the risk of bias related to random sequence generation (selection bias), allocation concealment (selection bias), blinding of participants and providers (performance bias), blinding of outcome assessors (detection bias), completeness of reporting outcome data (attrition bias), and selective outcome reporting (reporting bias) [31].

## Statistical analyses

We conducted pairwise meta-analyses of all direct comparisons to assess the statistical heterogeneity within each comparison. We then qualitatively examined the distribution of characteristics across studies in each network that may modify intervention effects to assess the transitivity assumption of NMA—that is, that participants hypothetically could be randomized to any interventions included in a network [32,33]. This transitivity assumption involves assuming that sets of studies comparing different interventions in a network are sufficiently similar to each other with respect to characteristics that moderate the relative effects of interventions, and this assumption leads to assessments of consistency of direct and indirect evidence within each network [33]. We assessed transitivity (similar distribution of potential effect modifiers across studies) by systematically tabulating and examining characteristics across trials [32]. Overall, we considered identified interventions to be comparable, as they are used in specialty care clinical settings as acute treatment for patients with comorbid alcohol use and depressive disorders [7,8]. The only major difference across trials that concerned us regarding the transitivity assumption of NMA involved the length of treatment, which ranged from 3 to 26 weeks. We consequently tabulated and compared length of treatment across intervention arms in each network (available in S2 Appendix), and we downgraded confidence in network estimates in which we judged the risk of intransitivity to be high as a result of considerable differences in treatment lengths.

We conducted NMA using random effects models in a frequentist framework with the net-meta package (version 0.9–8) in the R statistical environment [34]. We addressed within-study correlation of effects from multiarm trials through the netmeta procedures for reweighting all comparisons of each multiarm trial [35,36]. We assumed a constant heterogeneity variance across all comparisons in each network, defined via a generalized methods of moments estimate of the between-studies variance [37]. We assessed between-study clinical and methodological heterogeneity by examining study characteristics, between-study statistical heterogeneity for each pairwise comparison using the $I^2$ statistic, local inconsistency by splitting and comparing direct and indirect evidence [38], and global inconsistency using design-based decomposition of Cochran's Q and net heat plots [39].

We grouped outcome data into different follow-up periods: immediately postintervention, short-term follow-up (1 to 5 months postintervention), long-term follow-up (6 to 11 months postintervention), and very long-term follow-up (12+ months postintervention). For each combination of pairwise comparison, outcome, and time point, we used standardized mean differences (SMDs) for continuous outcomes and odds ratios (ORs) for dichotomous outcomes to estimate intervention effects. For consistency, we coded outcome data such that SMDs <0 and ORs < 1 are favorable, and we used established benchmarks for interpreting clinical effect sizes using SMDs and ORs, i.e., SMD ≤ −0.2 or OR ≥1.68 for a small clinical effect, SMD ≤ −0.5 or OR ≥3.47 for a medium clinical effect, and SMD ≤ −0.8 or OR ≥6.71 for a large clinical effect [40]. For each outcome and time point, we ranked interventions in order of effectiveness using *p* scores—a frequentist measure of the extent of certainty that an intervention is better than another intervention averaged over all competing interventions [41,42]. We conducted sensitivity analyses in which we excluded studies that evaluate pharmacological interventions that do not have legal approval to be prescribed in the United States, excluded studies with high risks of bias, and used alternative outcome data reported in included studies (e.g., studies that used multiple measures within a given outcome domain). In response to peer reviewer comments, we also conducted sensitivity analyses excluding studies prior to DSM-III (i.e., before 1980). We report results from the sensitivity analyses in the narrative only when we have high, moderate, or low confidence in an estimate that substantively changes the conclusions from the primary analysis (i.e., direction or size of the effect). Results of all sensitivity analyses can be found in the RMarkdown output file accompanying the manuscript (https://osf.io/bwyq8/). The actual RMarkdown file can be found in S1 Appendix.

## Rating confidence in effect estimates

We rated our confidence in each pairwise effect estimate and relative rankings of identified interventions using the Grading of Recommendations Assessment, Development, and Evaluation (GRADE) approach [43–46]. The traditional approach involves initially assigning a body of direct evidence of RCTs a rating of "high" confidence and then assessing 5 domains for possible downgrading of confidence by 1 or 2 levels. For limitations of included studies (none, serious, or very serious), we considered downgrading 1 level ("serious") when most information is from studies at moderate risk of bias and 2 levels ("very serious") from studies at high risk of bias. For indirectness (none, serious, or very serious), we considered downgrading 1 level ("serious") when some differences exist between the population, the intervention, or the outcomes measured in relevant research studies and those under consideration in our review and 2 levels ("very serious") when substantial differences exist. For inconsistency (none, serious, or very serious), we considered downgrading 1 level ("serious") when substantial heterogeneity existed or when only 2 studies provided information to a meta-analytic estimate and 2 levels ("very serious") when considerable heterogeneity existed or when only 1 study provided

information to a meta-analytic estimate. For imprecision (none, serious, or very serious), we considered downgrading 1 level ("serious") when the 95% CI included the null effect and 2 levels ("very serious") when the 95% CI included appreciable benefit or harm. For publication bias (suspected or undetected), we considered downgrading 1 level ("suspected") when evidence suggested a selective publication of study findings that likely substantially alters estimates of a non-null effect.

For pairwise estimates in an NMA, rating confidence in indirect evidence for an effect estimate involved taking the lowest confidence rating from effect estimates with a common comparator and assessing whether to downgrade for potential intransitivity. The process further involves (1) presenting the direct and indirect effect estimates for the pairwise comparison; (2) rating confidence in both estimates; (3) presenting the network estimate for the pairwise comparison; and (4) rating the confidence of the network estimate based on the ratings of the direct and indirect estimates, as well as an assessment of coherence [43,45]. Based on these assessments, we reported our confidence in each pairwise effect estimate using 1 of 4 categories [47,48]. "High" confidence indicates that we are very confident that there is (or is not) a non-null effect—that a pairwise effect estimate indicates that one intervention is beneficial over (superior to) another. "Moderate" confidence indicates that there likely is (not) a non-null effect. "Low" indicates that there may (not) be a non-null effect. "Very low" indicates that we are very uncertain about the existence (or not) of a non-null effect.

## Findings

All data underlying the findings in this review (including GRADE assessments of confidence in effect estimates) can be found in S2 Appendix.

### Study selection and characteristics

After identifying 5,452 citations for review, we excluded 4,758 citations during title and abstract screening, yielding 694 citations for full-text eligibility assessment. We excluded 596 citations at this stage, 16 citations related to ongoing studies or those awaiting a final assessment. We ultimately included 98 citations reporting 36 studies that randomized a total of 2,729 participants (see Fig 1).

A concise summary of study characteristics can be found in Tables 1 and 2. Studies were published from 1971 to 2019 and conducted in 13 countries. All studies randomized individual participants (rather than clusters of participants) to intervention groups. Aside from one 3-arm trial (3%) and one 4-arm trial (3%), most studies ($n$ = 34; 94%) randomized participants to either of 2 intervention groups. Only 4 studies (11%) met the minimum sample requirements of a reported power analysis. The majority of studies ($n$ = 28; 78%) involved only 1 site. The median study sample size was 50 participants (range, 14 to 350 participants). Regarding risk of bias (see Table 3), most studies did not report their random sequence generation ($n$ = 27; 75%) or allocation concealment ($n$ = 28, 78%) methods. Most studies evaluating pharmacological interventions ($n$ = 24, 86%) had low risk of performance bias related to blinding participants and providers. Twenty-two studies (61%) had low risk of detection bias related to blinding outcome assessors, while 18 studies (50%) had low risk of attrition bias related to completeness of outcome data at postintervention. Most studies had unclear ($n$ = 13; 36%) or high ($n$ = 18; 50%) risk of reporting bias related to selective reporting of outcome data. Ten studies (28%) reported at least 1 private source of funding with a potential conflict of interest.

Twenty-eight studies (78%) reported the stage in the clinical pathway and the type of clinical setting. Fifteen studies (42%) involved outpatient care, 9 (25%) inpatient care, and 4 (11%) inpatient followed by outpatient care. Seventeen studies (47%) took place in a treatment setting

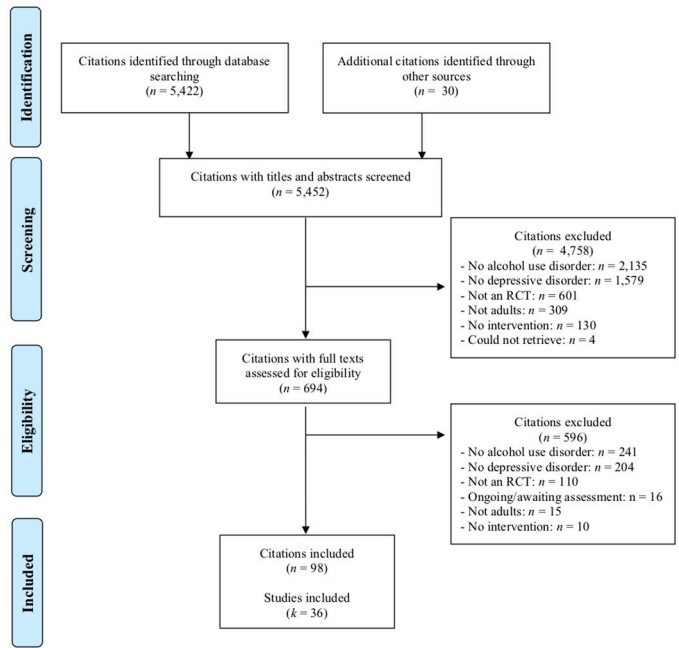

**Fig 1. Study flow diagram.** Note: A single study can have multiple publications and therefore citations. Therefore, in the "included" box in the flow diagram, we have noted how many studies are reported by included citations. RCT, randomized controlled trial.

primarily focused on alcohol or substance use, 1 (3%) in a treatment setting primarily focused on depressive or mental health disorders, and 10 (28%) in a dual-treatment setting. Participants in most studies had diagnoses of major depressive disorder ($n$ = 18; 50%) and alcohol dependence ($n$ = 20; 56%). The median average age was 42 years (range, 32 to 63), and the median percentage of female participants was 31% (range, 0% to 100%). Among the 20 studies (56%) that provided data on race/ethnicity, the median percentage of white participants was 76% (range, 47% to 100%). The available evidence included 14 pharmacological interventions, 4 psychological interventions, and 3 control interventions. All interventions were provided as adjunctive to treatment as usual (TAU) in the study setting.

## Network geometry

The available body of evidence contained 75 intervention arms in 3 overarching categories: pharmacological interventions ($n$ = 37), psychological interventions ($n$ = 10), and control interventions ($n$ = 28). Because we did not identify any studies directly comparing a pharmacological intervention with a psychological intervention, we analyzed 2 separate networks: one of pharmacological interventions (see Fig 2a) and another of psychological interventions (see Fig 2b). In the pharmacological network, the most common nodes were pharmacological placebos (21 trial groups; 685 participants), selective serotonin reuptake inhibitors (SSRIs; 12 trial groups; 611 participants), and tricyclic antidepressants (TCAs; 8 trial groups; 291 participants), with SSRIs versus pharmacological placebos as the most frequent direct comparison (8 comparisons). In the psychological network, the most common nodes were self-management support (4 trial groups; 97 participants), cognitive behavioral therapies (CBTs; 3 trial groups; 230 participants), psychological placebos (3 trial groups; 55 participants), and no additional treatment (3 trial groups; 238 participants), with self-management support versus psychological placebos as the most frequent direct comparison (3 comparisons).

**Table 1. Summary of studies included in the pharmacological intervention network.**

| Study | Country | N | Age | Female | Depression | Alcohol | Stage | Setting | Sites | Intervention 1 | Intervention 2 | Weeks | Cointervention |
|---|---|---|---|---|---|---|---|---|---|---|---|---|---|
| **SSRI vs. placebo** | | | | | | | | | | | | | |
| Adamson (2015) | New Zealand | 138 | 44 | 59% | Major depressive episode (DSM-IV) | Alcohol dependence (DSM-IV) | Outpatient | SUD | 8 | Citalopram | Pharmacologic placebo | 12 | Pharmacotherapy, outpatient program |
| Cornelius and colleagues (1997) | US | 51 | 35 | 49% | Major depressive disorder (DSM-III) | Alcohol dependence (DSM-III) | Inpatient + outpatient | Both | 1 | Fluoxetine | Pharmacologic placebo | 12 | Inpatient + outpatient program |
| Gual (2003) | Spain | 83 | 47 | 53% | Major depressive disorder and/or dysthymic disorder (DSM-IV/ICD-10) | Alcohol dependence (DSM-IV/ICD-10) | Outpatient | SUD | 1 | Sertraline | Pharmacologic placebo | 24 | Outpatient program |
| Kranzler and colleagues (2006) | US | 345 | 43 | 36% | Major depressive disorder (DSM-IV) | Alcohol dependence (DSM-IV) | NR | NR | 13 | Sertraline | Pharmacologic placebo | 10 | Outpatient program |
| Krupitsky and colleagues (2013) | Russia | 60 | 42 | 22% | Depressive episode (ICD-10) | Alcohol dependence (ICD-10) | Inpatient | SUD | 1 | Escitalopram | Pharmacologic placebo | 13 | Outpatient program |
| Moak and colleagues (2003) | US | 82 | 42 | 39% | Major depressive episode or dysthymic disorder (DSM-III) | Alcohol abuse or dependence (DSM-III) | NR | NR | 1 | Sertraline | Pharmacologic placebo | 12 | Psychotherapy |
| Pettinati and colleagues (2010) | US | 170 | 43 | 38% | Major depressive disorder (DSM-IV) | Alcohol dependence (DSM-IV) | Outpatient | SUD | 1 | Sertraline | Pharmacologic placebo | 14 | Psychotherapy |
| Roy (1998) | US | 36 | 41 | 8% | Major depressive episode (DSM-III) | Alcohol dependence (DSM-III) | Inpatient + outpatient | SUD | 1 | Sertraline | Pharmacologic placebo | 6 | Inpatient + outpatient program |
| **SSRI vs. opioid antagonist + SSRI** | | | | | | | | | | | | | |
| Pettinati and colleagues (2010) | US | 170 | 43 | 38% | Major depressive disorder (DSM-IV) | Alcohol dependence (DSM-IV) | Outpatient | SUD | 1 | Sertraline | Naltrexone + sertraline | 14 | Psychotherapy |
| Salloum (2007) | US | 106 | NR | 46% | Major depressive disorder (DSM-IV) | Alcohol dependence (DSM-IV) | NR | NR | 1 | Fluoxetine | Naltrexone + fluoxetine | 26 | Psychotherapy |
| **SSRI vs. AAP + SSRI** | | | | | | | | | | | | | |
| Han and colleagues (2013) | South Korea | 35 | 40 | 34% | Major depressive disorder (DSM-IV) | Alcohol dependence (DSM-IV) | Inpatient | Both | 2 | Escitalopram | Aripiprazole + escitalopram | 6 | Detoxification, inpatient program |
| **SSRI vs. NMDA antagonist** | | | | | | | | | | | | | |
| Muhonen and colleagues (2008) | Finland | 80 | 48 | 45% | Major depressive disorder (DSM-IV) | Alcohol dependence (DSM-IV) | Outpatient | SUD | 3 | Escitalopram | Memantine | 26 | Outpatient program |
| **SSRI vs. opioid antagonist** | | | | | | | | | | | | | |
| Pettinati and colleagues (2010) | US | 170 | 43 | 38% | Major depressive disorder (DSM-IV) | Alcohol dependence (DSM-IV) | Outpatient | SUD | 1 | Sertraline | Naltrexone | 14 | Psychotherapy |
| **SSRI vs. TCA** | | | | | | | | | | | | | |
| Cocchi (1997) | Italy | 122 | 42 | 22% | "Depression" | "Alcoholic" | Inpatient | SUD | 1 | Paroxetine | Amitryptiline | 3 to 4 | Detoxification |
| **TCA vs. placebo** | | | | | | | | | | | | | |

*(Continued)*

**Table 1.** (Continued)

| Study | Country | N | Age | Female | Depression | Alcohol | Stage | Setting | Sites | Intervention 1 | Intervention 2 | Weeks | Cointervention |
|---|---|---|---|---|---|---|---|---|---|---|---|---|---|
| Butterworth (1971) | US | 40 | 23 to 60 | 0% | Depression (LRDR of 10 + or clinical impression) | "Alcoholic" | Inpatient | SUD | 1 | Imipramine | Pharmacologic placebo | 3 | Detoxification, inpatient program |
| Mason and colleagues (1996) | US | 28 | 39 | 14% | Major depressive disorder (DSM-III) | Alcohol dependence (DSM-III) | Outpatient | SUD | 2 | Desipramine | Pharmacologic placebo | 26 | Psychotherapy, self-help group |
| McGrath and colleagues (1996) | US | 69 | 37 | 51% | Major depressive, dysthymic, depressive disorder not otherwise specified (DSM-III) | Alcohol abuse or dependence (DSM-III) | Outpatient | MH | 1 | Imipramine | Pharmacologic placebo | 12 | Outpatient program |
| *TCA vs. MRI* | | | | | | | | | | | | | |
| Mielke and Gallant (1978) | US | 20 | 37 | NR | Major depressive disorder (DSM-II) | "Alcoholism" | Inpatient | SUD | 1 | Imipramine | AHR-1118 | 4 | Inpatient program |
| *TCA vs. TCA* | | | | | | | | | | | | | |
| Loo and colleagues (1988) | France | 129 | 38 | 14% | Major depressive episode or dysthymic disorder (DSM-III) | Alcohol abuse or dependence (DSM-III) | NR | NR | 7 | Tianeptine | Amitriptyline | 4 to 8 | Pharmacotherapy |
| *TCA vs. TeCA* | | | | | | | | | | | | | |
| Altintoprak and colleagues (2008) | Turkey | 44 | 4 | 8% | Major depressive disorder (DSM-IV) | Alcohol dependence (DSM-IV) | Inpatient | SUD | 1 | Amitriptyline | Mirtazapine | 8 | Detoxification |
| *TeCA vs. placebo* | | | | | | | | | | | | | |
| Cornelius and colleagues (2016) | US | 14 | 41 | 29% | Major depressive disorder (DSM-IV) | Abuse or dependence (DSM-IV) | Outpatient | Both | 1 | Mirtazapine | Pharmacologic placebo | 12 | Psychotherapy |
| McLean and colleagues (1986) | United Kingdom | 35 | 37 | 31% | Depression (HDRS of 17+) | "Alcohol dependence" | Inpatient | SUD | 1 | Mianserin | Pharmacologic placebo | 4 | Inpatient program |
| *Opioid antagonist vs. placebo* | | | | | | | | | | | | | |
| Oslin (2005) | US | 74 | 63 | 20% | Major depressive disorder (DSM-IV) | Alcohol dependence (DSM-IV) | NR | NR | 1 | Naltrexone | Pharmacologic placebo | 12 | Psychotherapy |
| Pettinati and colleagues (2010) | US | 170 | 43 | 38% | Major depressive disorder (DSM-IV) | Alcohol dependence (DSM-IV) | Outpatient | SUD | 1 | Naltrexone | Pharmacologic placebo | 14 | Psychotherapy |
| *Opioid antagonist vs. opioid antagonist + SSRI* | | | | | | | | | | | | | |
| Pettinati and colleagues (2010) | US | 170 | 43 | 38% | Major depressive disorder (DSM-IV) | Alcohol dependence (DSM-IV) | Outpatient | SUD | 1 | Naltrexone + sertraline | Naltrexone | 14 | Psychotherapy |
| *SARI vs. placebo* | | | | | | | | | | | | | |
| Hernandez-Avila and colleagues (2004) | US | 41 | 43 | 51% | Major depressive disorder (DSM-IV) | Alcohol dependence (DSM-IV) | Outpatient | Both | 1 | Nefazodone | Pharmacologic placebo | 10 | Psychotherapy |
| Roy-Byrne and colleagues (2000) | US | 64 | 40 | 55% | Major depressive disorder (DSM-III) | Alcohol dependence (DSM-III) | Outpatient | Both | 1 | Nefazodone | Pharmacologic placebo | 12 | Psychotherapy |
| *Opioid antagonist + SSRI vs. placebo* | | | | | | | | | | | | | |

(*Continued*)

**Table 1.** (Continued)

| Study | Country | N | Age | Female | Depression | Alcohol | Stage | Setting | Sites | Intervention 1 | Intervention 2 | Weeks | Cointervention |
|---|---|---|---|---|---|---|---|---|---|---|---|---|---|
| Pettinati and colleagues (2010) | US | 170 | 43 | 38% | Major depressive disorder (DSM-IV) | Alcohol dependence (DSM-IV) | Outpatient | SUD | 1 | Naltrexone + sertraline | Pharmacologic placebo | 14 | Psychotherapy |
| *AAP vs. placebo* | | | | | | | | | | | | | |
| Golik-Gruber and colleagues (2003) | Croatia | 40 | NR | 0% | "Depression" | "Alcohol addiction" | Inpatient | SUD | 1 | Sulpride | Pharmacologic placebo | 3 | Outpatient program |
| ***Glutamatergic antagonist vs. placebo*** | | | | | | | | | | | | | |
| Witte and colleagues (2012) | US | 23 | 46 | 43% | Major depressive disorder (DSM-IV) | Abuse or dependence (DSM-IV) | NR | NR | 1 | Acamprosate | Pharmacologic placebo | 12 | Outpatient program |
| *nAChRs vs. placebo* | | | | | | | | | | | | | |
| Ralevski and colleagues (2013) | US | 21 | 50 | 29% | Major depressive disorder (DSM-IV) | Alcohol dependence (DSM-IV) | NR | NR | 1 | Mecamylamine | Pharmacologic placebo | 12 | Outpatient program |
| *NRI vs. placebo* | | | | | | | | | | | | | |
| Altamura and colleagues (1990) | NR | 30 | 45 | 20% | Dysthymic disorder (DSM-III) | Alcohol dependence (DSM-III) | NR | NR | 1 | Viloxazine | Pharmacologic placebo | 12 | Inpatient + outpatient program |
| *TCA + sedative vs. placebo* | | | | | | | | | | | | | |
| Shaw and colleagues (1975) | US | 30 | 32 | 0% | Depression (BDI, MMPI, ZDS) | Alcoholism (National Council on Alcoholism) | Inpatient + outpatient | SUD | 1 | Chlordiazepoxide + imipramine | Pharmacologic placebo | 4 | Inpatient + outpatient program |

AAP, atypical antipsychotic; BDI, Beck Depression Inventory; DSM, Diagnostic and Statistical Manual of Mental Disorders; HDRS, Hamilton Depression Rating Scale; ICD, International Classification of Diseases; LRDR, Lehmann-Rockliff Depression Rating; MH, mental health; MMPI, Minnesota Multiphasic Personality Inventory; MRI, monoamine reuptake inhibitor; nAChR, nonselective noncompetitive antagonists of the nicotinic acetylcholine receptor; NMDA, N-methyl-D-aspartate; NR, not reported; NRI, norepinephrine reuptake inhibitor; SARI, serotonin antagonist and reuptake inhibitor; SSRI, selective serotonin reuptake inhibitor; SUD, substance use disorder; TCA, tricyclic antidepressant; TeCA, tetracyclic antidepressant; ZDS, Zung's Self-Rating Depression Scale.

**Table 2. Summary of studies included in the psychological intervention network.**

| Study | Country | N | Age | Female | Depression | Alcohol | Stage | Setting | Sites | Intervention 1 | Intervention 2 | Weeks | Cointervention |
|---|---|---|---|---|---|---|---|---|---|---|---|---|---|
| **CBT vs. no additional treatment** | | | | | | | | | | | | | |
| Thapinta and colleagues (2014) | Thailand | 80 | 45 | 18% | Mild Depression or Moderate Depression (9Q of 7 to 18) | "Alcohol Dependence" | Outpatient | Both | 5 | Brief CBT | No additional treatment | 3 | Psychotherapy, pharmacotherapy |
| Thapinta and colleagues (2017) | Thailand | 350 | 39 | 12% | Mild Depression (PHQ-9 of 5 to 8) | Alcohol Dependence (DSM-IV) | Outpatient | Both | 5 | CBT Self Help Book | No additional treatment | 1 | Psychotherapy |
| **CBT vs. placebo** | | | | | | | | | | | | | |
| Petersen and Zettle (2009) | US | 30 | 38 | 50% | Major Depressive Disorder (DSM-IV) | Alcohol Abuse or Dependence (DSM-IV) | Inpatient | SUD | 1 | Acceptance and commitment therapy | Psychological placebo | 3 to 4 | Inpatient program |
| **IPT vs. no additional treatment** | | | | | | | | | | | | | |
| Holzhauer and colleagues (2017) | US | 48 | 37 | 100% | Major Depressive Disorder (DSM-IV) | Alcohol Dependence (DSM-IV) | Outpatient | SUD | 1 | Interpersonal therapy | No additional treatment | 16 | Outpatient program |
| **IPT vs. SP** | | | | | | | | | | | | | |
| Markowitz and colleagues (2008) | US | 26 | 38 | 31% | Dysthymic Disorder (DSM-IV) | Alcohol Abuse or Dependence (DSM-IV) | Outpatient | Both | 1 | Interpersonal therapy for dysthymic disorder | Brief supportive psychotherapy | 16 | Pharmacotherapy, self-help group |
| **Self-management support vs. no additional treatment** | | | | | | | | | | | | | |
| O'Reilly and colleagues (2019) | Ireland | 95 | 48 | 46% | Major Depressive Episode (DSM-IV) | Alcohol Dependence (DSM-IV) | Outpatient | Both | 1 | Supportive text messaging | No additional treatment | 26 | Outpatient aftercare program (support group) |
| **Self-management support vs. placebo** | | | | | | | | | | | | | |
| Agyapong and colleagues (2012) | Ireland | 54 | 49 | 54% | Major Depressive Disorder (DSM-IV) | Alcohol Abuse or Dependence (DSM-IV) | Outpatient | Both | 1 | Supportive text messaging | Psychological placebo | 13 | Inpatient + outpatient program |
| Zielinski (1979) | US | 36 | 40 | 25% | Depression (BDI, MMPI, Zung's Self-Rating Depression Scale) | "Alcoholic" | Inpatient + Outpatient | SUD | 1 | Activity level monitoring + social skills training | Psychological placebo | 13 | Inpatient + outpatient program |
| Zielinski (1979) | US | 36 | 40 | 25% | Depression (BDI, MMPI, Zung's Self-Rating Depression Scale) | "Alcoholic" | Inpatient + Outpatient | SUD | 1 | Activity level monitoring | Psychological placebo | 13 | Inpatient + outpatient program |

CBT, cognitive behavioral therapy; DSM, Diagnostic and Statistical Manual of Mental Disorders; IPT, interpersonal therapy; PHQ, Patient Health Questionnaire; SP, supportive psychotherapy; SUD, substance use disorder.

**Table 3. Summary of risks of bias across studies.**

| Study | Random sequence generation | Allocation concealment | Blinding participants | Blinding providers | Blinding assessors | Completeness of outcome data | Selective outcome reporting | Funder |
|---|---|---|---|---|---|---|---|---|
| Adamson (2015) | Low | Low | Low | Low | Low | Low | Low | Public |
| Agyapong (2012) | Low | Unclear | High | Low | High | Low | Low | Public |
| Altamura (1990) | Unclear | Unclear | Low | Low | Unclear | Low | High* | Unclear |
| Altintoprak (2008) | Unclear | Unclear | Low | Low | Unclear | Unclear | High* | Unclear |
| Butterworth (1971) | Unclear | Unclear | Low | Low | Unclear | Low | Unclear | Unclear |
| Cocchi (1997) | Unclear | Unclear | Unclear | Unclear | Unclear | Low | Unclear | Unclear |
| Cornelius (1997) | Unclear | Low | Low | Low | Low | Low | Unclear | Public |
| Cornelius (2016) | Unclear | Low | Low | Low | Unclear | Low | Low | Public |
| Golik-Gruber (2003) | Unclear | Unclear | High | High | High | Low | Unclear | Unclear |
| Gual (2003) | Unclear | Unclear | Low | Low | Unclear | High | High* | Unclear |
| Han (2013) | Unclear | Unclear | Low | Low | Unclear | High* | Unclear | Some private |
| Hernandez-Avila (2004) | Low | Unclear | High | Low | Low | Low | Unclear | Some private |
| Holzhauer (2017) | Unclear | Unclear | High | High | Unclear | Low | High* | Public |
| Kranzler (2006) | Low | Unclear | Low | Low | Unclear | High | High* | Some private |
| Krupitsky (2013) | Low | Low | Low | Low | Low | Low | High* | Unclear |
| Loo (1988) | Unclear | Unclear | Low | Low | Unclear | Low | Unclear | Unclear |
| Markowitz (2008) | Low | Unclear | High | High | Low | High | Unclear | Public |
| Mason (1996) | Unclear | Unclear | Low | Low | Unclear | High | High* | Public |
| McGrath (1996) | Unclear | Unclear | Low | Low | Low | High | Unclear | Some private |
| McLean (1986) | Unclear | Unclear | Low | Low | Low | Low | Unclear | Unclear |
| Mielke (1978) | Unclear | Unclear | Low | Low | Unclear | Low | High* | Public |
| Moak (2003) | Unclear | Unclear | Low | Low | Unclear | Unclear | High* | Some private |
| Muhonen (2008) | Low | Low | Low | Low | Unclear | High | Low | Some private |
| O'Reilly (2019) | Low | Low | High | High | Low | High | High* | Public |
| Oslin (2005) | Unclear | Unclear | Unclear | Unclear | Unclear | High | High* | Some private |
| Petersen (2009) | Low | Low | High | High | Low | High | High* | Unclear |
| Pettinati (2010) | Unclear | Unclear | Low | Low | Unclear | High* | High* | Some private |
| Ralevski (2013) | Unclear | Unclear | Low | Low | Low | Low | High* | Public |
| Roy (1998) | Unclear | Unclear | Low | Low | Low | High* | High* | Unclear |
| Roy-Byrne (2000) | Unclear | Unclear | Low | Low | Low | High | High* | Some private |
| Salloum (2007) | Unclear | Unclear | Low | Low | Unclear | Unclear | High* | Public |
| Shaw (1975) | Unclear | Unclear | Low | Low | Unclear | Low | High* | Unclear |
| Thapinta (2014) | Unclear | Unclear | High | High | Unclear | High | Unclear | Public |
| Thapinta (2017) | Unclear | Unclear | High | High | Low | Low | Unclear | Public |
| Witte (2012) | Unclear | Unclear | Low | Low | Unclear | High | Low | Some private |
| Zielinski (1979) | Unclear | Unclear | High | High | Unclear | Low | Unclear | Unclear |

* Indicates that the high risk of bias is for some but not all outcomes.

Figure 2a. Pharmacological Network Structure by Intervention Class

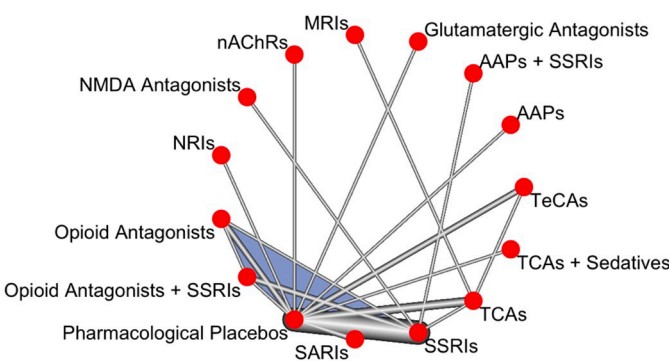

Figure 2b. Psychological Network Structure by Intervention Class

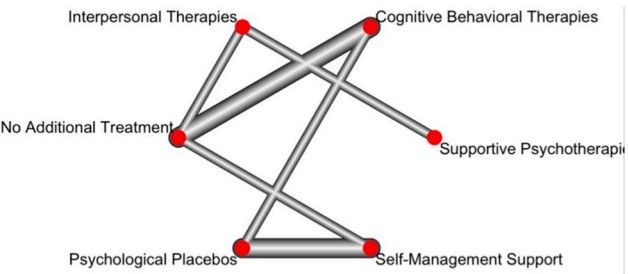

**Fig 2. Network structure. (a)** Pharmacological network structure by intervention class. **(b)** Psychological network structure by intervention class. Notes: The size of the width of each edge (line) is based on the number of direct comparisons between the 2 connected interventions. The shaded area indicates a "closed" loop (i.e., there is at least 1 study that compares one alternative intervention to an intervention with another alternative to that intervention), allowing the comparison of effect estimates from direct evidence with effect estimates from indirect evidence. AAP, atypical antipsychotic; MRI, monoamine reuptake inhibitor; nAChR, nonselective noncompetitive antagonists of the nicotinic acetylcholine receptor; NRI, norepinephrine reuptake inhibitor; SARI, serotonin antagonist and reuptake inhibitor; SSRI, selective serotonin reuptake inhibitor; TCA, tricyclic antidepressant; TeCA, tetracyclic antidepressant.

## Network meta-analyses

This body of evidence predominantly consists of psychometrically validated questionnaires measuring constructs immediately at postintervention. A summary of outcome data across studies can be found in Table 4, while a summary of findings from the network meta-analyses (including effect estimates, intervention rankings, and confidence in the evidence) can be found in Tables 5 and 6.

**Remission from depression.** Eighteen pharmacological intervention studies (64%) reported data on 12 intervention classes, while 2 psychological intervention studies (25%) reported data on 4 intervention classes. We did not detect significant heterogeneity in the pharmacologic network globally ($Q(3) = 4.89$, $p = 0.18$), and hotspots of inconsistency were absent from the net heat plot. Based on confidence ratings using the GRADE approach, we have very low confidence in all effect estimates, meaning we are very uncertain about the existence (or not) of a non-null effect based on the available evidence. Sensitivity analyses did not substantively differ from the primary analyses for remission from depression.

**Remission from alcohol use.** Seventeen pharmacological intervention studies (61%) reported data on 9 intervention classes, while 3 psychological intervention studies (38%) reported data on 5 intervention classes. We did not detect significant heterogeneity in the pharmacologic network globally ($Q(2) = 0.73$, $p = 0.69$), and hotspots of inconsistency were

**Table 4. Summary of outcome data across studies.**

| Study | Remission from depression | Remission from alcohol use | Depressive symptoms | Alcohol use | Heavy drinking | Withdrawal and craving symptoms | Health-related quality of life | Functional status | Adverse events |
|---|---|---|---|---|---|---|---|---|---|
| Adamson and colleagues (2015) | Proportion with MADRS <10 at postintervention | Mean percentage days abstinent at postintervention | Mean MADRS at postintervention | Mean drinks per drinking day at postintervention | Mean percentage of days heavy drinking at postintervention | Mean LDQ at postintervention | — | — | Proportion self-reporting at least 1 adverse event at postintervention |
| Agyapong and colleagues (2012) | — | Proportion abstinent at immediately and 3 months postintervention | Mean BDI at immediately and 3 months postintervention | Mean units of alcohol per drinking day at immediately and 3 months postintervention | — | Mean OCDS at immediately and 3 months postintervention | — | Mean GAF at immediately and 3 months postintervention | — |
| Altamura and colleagues (1990) | — | — | — | — | — | — | — | — | — |
| Altintoprak and colleagues (2008) | — | — | — | — | — | Mean ACS at postintervention | — | — | — |
| Butterworth (1971) | — | — | Mean LRDR at postintervention | — | — | — | — | Proportion with "improvement in global response" at postintervention | Proportion with at least 1 physician-recorded adverse event at postintervention |
| Cocchi (1997) | Proportion "not depressed" using ZDS at postintervention | — | Mean ZDS at postintervention | — | — | — | — | — | — |
| Cornelius and colleagues (1997) | — | Proportion abstinent at postintervention Mean number of days abstinent at 9 months postintervention | Mean HDRS at immediately and 9 months postintervention | Mean drinks per drinking day at postintervention Number drinks in the past week at 9 months postintervention | Cumulative number of days heavy drinking at postintervention | — | — | Mean GAS at immediately and 9 months postintervention | Proportion with at least 1 side effect at immediately and 9 months postintervention |
| Cornelius and colleagues (2016) | — | — | Mean BDI at postintervention | Mean drinks per drinking day at postintervention | Mean number of days heavy drinking per week at postintervention | Mean OCDS at postintervention | — | — | — |
| Golik-Gruber and colleagues (2003) | Proportion with depression (BDI) at postintervention | — | — | — | — | — | — | — | — |
| Gual and colleagues (2003) | Proportion with ≥50% reduction on MADRS at postintervention | Proportion who did not relapse at postintervention | — | — | — | — | — | — | — |

(*Continued*)

**Table 4.** (Continued)

| Study | Remission from depression | Remission from alcohol use | Depressive symptoms | Alcohol use | Heavy drinking | Withdrawal and craving symptoms | Health-related quality of life | Functional status | Adverse events |
|---|---|---|---|---|---|---|---|---|---|
| Han and colleagues (2013) | Proportion with ≥50% reduction on BDI at postintervention | Proportion who did not relapse at postintervention | Mean BDI at postintervention | — | — | Mean KAUQ at postintervention | — | Mean CGI severity at postintervention | Proportion who dropped out because of adverse events at postintervention |
| Hernandez-Avila and colleagues (2004) | — | Proportion abstinent at postintervention | Mean HDRS at postintervention | Mean drinks per drinking day at postintervention | Mean number of days heavy drinking per week at postintervention | — | — | — | Mean SAFTEE score at postintervention |
| Holzhauer and Gamble (2017) | — | — | — | — | — | — | — | — | — |
| Kranzler and colleagues (2006) | Proportion with ≥50% reduction on HDRS at postintervention | Mean percentage days abstinent at postintervention | Mean HDRS at postintervention | — | — | — | — | — | Proportion with a treatment emergent adverse event at postintervention |
| Krupitsky and colleagues (2013) | — | Mean number of days remission at postintervention | Mean HDRS at postintervention | — | Mean gamma-glutamyltransferase activity at postintervention | Mean OCDS at postintervention | — | Proportion with considerable or very considerable improvement on CGI at postintervention | Proportion with at least 1 adverse event at postintervention |
| Loo and colleagues (1988) | Proportion with ≥50% reduction on MADRS at postintervention | — | — | — | — | — | — | — | Proportion who discontinued treatment at postintervention |
| Markowitz and colleagues (2008) | Proportion with "remission" at postintervention | Mean percentage days abstinent at postintervention | Mean HDRS at postintervention | — | — | — | — | — | — |
| Mason and colleagues (1996) | Proportion with "depression response" at postintervention | Proportion who did not relapse at postintervention | Mean HDRS at postintervention | — | — | — | — | — | Proportion who dropped out because of adverse events at postintervention |
| McGrath and colleagues (1996) | Proportion with ≥50% reduction on HDRS at postintervention | Proportion with past week abstinence at postintervention | Mean HDRS at postintervention | Mean drinks per drinking day at postintervention | Mean percentage of days heavy drinking in the past week at postintervention | — | — | Proportion much improved or better on CGI at postintervention | Proportion who dropped out because of adverse events at postintervention |
| McLean and colleagues (1986) | Proportion with HDRS <17 at postintervention | — | Mean HDRS at postintervention | — | — | — | — | — | Proportion who experienced transient drowsiness at postintervention |

(Continued)

**Table 4.** (Continued)

| Study | Remission from depression | Remission from alcohol use | Depressive symptoms | Alcohol use | Heavy drinking | Withdrawal and craving symptoms | Health-related quality of life | Functional status | Adverse events |
|---|---|---|---|---|---|---|---|---|---|
| Mielke and Gallant (1978) | Proportion with moderate/marked improvement on the CGI at postintervention | — | — | — | — | — | — | — | Proportion with at least 1 side effect at postintervention |
| Moak and colleagues (2003) | Proportion with ≥50% reduction on BDI at postintervention | Mean percentage days abstinent at postintervention | Mean HDRS at postintervention | Mean drinks per drinking day at postintervention | — | — | — | — | Proportion with at least 1 adverse event at postintervention |
| Muhonen and colleagues (2008) | Proportion with MADRS <12 at postintervention | Mean AUDIT at postintervention | Mean MADRS at postintervention | Mean daily grams alcohol at postintervention | Mean number of days heavy drinking at postintervention | Mean OCDS at postintervention | Mean VAS at postintervention | Mean SOFAS at postintervention | Proportion with at least 1 adverse event at postintervention |
| O'Reilly and colleagues (2019) | — | Proportion who have consumed any alcohol immediately and at 6 months postintervention | BDI-II immediately and at 6 months postintervention | Mean (1) days drinking in past 3 months and (2) units of alcohol per drinking day immediately and at 6 months postintervention | — | Mean OCDS immediately and at 6 months postintervention | — | — | — |
| Oslin (2005) | Proportion with HDRS <10 at postintervention | Proportion abstinent at postintervention | — | — | Proportion relapsed to heavy drinking at postintervention | — | — | — | — |
| Petersen and Zettle (2009) | Proportion with HDRS <14 at postintervention | — | Mean HDRS at postintervention | — | — | — | — | — | — |
| Pettinati and colleagues (2010) | Proportion "not depressed" at postintervention | Proportion abstinent at postintervention | Mean HDRS at postintervention | — | Mean days to relapse to heavy drinking at postintervention | — | — | — | Proportion who discontinued treatment because of adverse events at postintervention |
| Ralevski and colleagues (2013) | — | — | Mean HDRS at postintervention | — | — | — | — | — | Proportion who experienced at least 1 medical or psychiatric adverse event at postintervention |
| Roy (1998) | Proportion with ≥50% reduction on HDRS at postintervention | Proportion who did not relapse at postintervention | Mean HDRS at postintervention | — | — | — | — | Proportion very much improved on CGI at postintervention | — |

(*Continued*)

**Table 4.** (Continued)

| Study | Remission from depression | Remission from alcohol use | Depressive symptoms | Alcohol use | Heavy drinking | Withdrawal and craving symptoms | Health-related quality of life | Functional status | Adverse events |
|---|---|---|---|---|---|---|---|---|---|
| Roy-Byrne (2000) | Proportion with HDRS <8 at postintervention | Proportion abstinent at postintervention | Mean HDRS at postintervention | Mean drinks per drinking day at postintervention | — | Mean VAS at postintervention | — | Proportion very much improved or much better on CGI at postintervention | Mean number of adverse events per participant at postintervention |
| Salloum (2007) | — | — | — | — | — | — | — | — | — |
| Shaw and colleagues (1975) | — | — | — | — | — | — | — | — | — |
| Thapinta and colleagues (2014) | — | — | Mean 9Q at immediately and 1 month postintervention | — | — | — | — | — | — |
| Thapinta and colleagues (2017) | — | — | Mean PHQ-9 at immediately, 1 month, and 6 months postintervention | Mean cubic cm of alcohol per day at immediately, 3 months, and 6 months postintervention | — | — | — | — | — |
| Witte (2012) | Proportion with "remission" at postintervention | Mean percentage days abstinent at postintervention | Mean HDRS at postintervention | Mean drinks per drinking day at postintervention | — | Mean OCDS at postintervention | Mean Q-LESQ at postintervention | Mean CGI improvement at postintervention | Proportion who experienced at least 1 adverse event at postintervention |
| Zielinski (1979) | — | Proportion who did not relapse at 6 and 12 months postintervention | — | | — | — | — | — | — |

ACS, Alcohol Craving Scale; AUDIT, Alcohol Use Disorder Identification Test; BDI, Beck Depression Inventory; CGI, Clinical Global Impression scale; cm, centimeters; GAF, Global Assessment of Function; GAS, Global Assessment Scale; GGT, Gamma-Glutamiltransferase; HDRS, Hamilton Depression Rating Scale; KAUQ, Korean Alcohol Urge Questionnaire; LDQ, Leeds Dependence Questionnaire; LRDR, Lehmann-Rockliff Depression Rating; MADRS, Montgomery-Åsberg Depression Rating Scale; OCDS, Obsessive Compulsive Drinking Scale; PHQ, Patient Health Questionnaire; Q-LESQ, Quality of Life Enjoyment and Satisfaction Questionnaire; SAFTEE, Systematic Assessment for Treatment of Emergent Events; SOFAS, Social and Occupational Functioning Assessment Scale; VAS, Visual Analog Scale; ZDS, Zung Depression Scale.

**Table 5. Summary of findings table for the pharmacological intervention network.**

| *Remission from depression at postintervention (tau$^2$ = 0.4764; I$^2$ = 68.7%)* | | | |
|---|---|---|---|
| **Intervention class** | **OR (95% CI)** | **Confidence in non-null effect** | **Ranking (p score)** |
| Opioid antagonist + SSRI | 0.22 (0.04 to 1.07) | Very low | 1 (0.81) |
| SARI | 0.21 (0.03 to 1.30) | Very low | 2 (0.80) |
| TCA | 0.39 (0.13 to 1.19) | Very low | 3 (0.67) |
| AAP + SSRI | 0.55 (0.07 to 4.63) | Very low | 4 (0.52) |
| AAP | 0.58 (0.08 to 4.25) | Very low | 5 (0.50) |
| Opioid antagonist | 0.66 (0.22 to 2.01) | Very low | 6 (0.46) |
| TeCA | 0.77 (0.11 to 5.46) | Very low | 7 (0.42) |
| SSRI | 0.75 (0.42 to 1.36) | Very low | 8 (0.41) |
| Glutamatergic antagonist | 0.80 (0.09 to 6.92) | Very low | 9 (0.41) |
| MRI | 0.92 (0.07 to 11.58) | Very low | 10 (0.38) |
| NMDA antagonist | 0.92 (0.15 to 5.84) | Very low | 11 (0.36) |
| Pharmacological placebo | — | — | 12 (0.26) |
| *Remission from alcohol use at postintervention (tau$^2$ = 0.0365; I$^2$ = 39.5%)* | | | |
| **Intervention class** | **OR (95% CI)** | **Confidence in non-null effect** | **Ranking (p score)** |
| Opioid antagonist + SSRI | 0.34 (0.15 to 0.79) | Very low | 1 (0.89) |
| TCA | 0.56 (0.19 to 1.68) | Very low | 2 (0.69) |
| NMDA antagonist | 0.59 (0.24 to 1.47) | Very low | 3 (0.68) |
| AAP + SSRI | 0.57 (0.08 to 3.92) | Very low | 4 (0.63) |
| SARI | 0.76 (0.27 to 2.10) | Very low | 5 (0.55) |
| Pharmacological placebo | — | — | 6 (0.40) |
| SSRI | 1.06 (0.84 to 1.34) | Very low | 7 (0.33) |
| Opioid antagonist | 1.41 (0.71 to 2.77) | Very low | 8 (0.19) |
| Glutamatergic antagonist | 2.10 (0.45 to 9.84) | Very low | 9 (0.15) |
| *Remission from alcohol use at long-term follow-up (tau$^2$ = 0; I$^2$ = 0%)* | | | |
| **Intervention class** | **OR (95% CI)** | **Confidence in non-null effect** | **Ranking (p score)** |
| SSRI | 0.26 (0.07 to 0.97) | Very low | 1 (N/A) |
| Pharmacological placebo | — | — | 2 (N/A) |
| *Depressive symptoms at postintervention (tau$^2$ = 0.0433; I$^2$ = 42.2%)* | | | |
| **Intervention class** | **SMD (95% CI)** | **Confidence in non-null effect** | **Ranking (p score)** |
| Opioid antagonist + SSRI | −0.66 (−1.27 to −0.06) | Very low | 1 (0.85) |
| Opioid antagonist | −0.49 (−1.10 to 0.12) | Very low | 2 (0.73) |
| TCA | −0.37 (−0.72 to −0.02) | Low | 3 (0.66) |
| SARI | −0.31 (−0.80 to 0.19) | Very low | 4 (0.59) |
| Glutamatergic antagonist | −0.33 (−1.25 to 0.59) | Very low | 5 (0.58) |
| AAP + SSRI | −0.21 (−1.06 to 0.63) | Very low | 6 (0.49) |
| SSRI | −0.14 (−0.35 to 0.07) | Very low | 7 (0.42) |
| TeCA | −0.12 (−0.78 to 0.53) | Very low | 8 (0.42) |
| NMDA antagonist | 0.03 (−0.67 to 0.74) | Very low | 9 (0.29) |
| Pharmacological placebo | — | — | 10 (0.24) |
| nAChRs | 0.20 (−0.75 to 1.15) | Very low | 11 (0.22) |
| *Depressive symptoms at long-term follow-up (tau$^2$ = 0; I$^2$ = 0%)* | | | |
| **Intervention class** | **SMD (95% CI)** | **Confidence in non-null effect** | **Ranking (p score)** |
| SSRI | −0.80 (−1.53 to −0.07) | Very low | 1 (N/A) |
| Pharmacological placebo | — | — | 2 (N/A) |
| *Withdrawal/craving symptoms at postintervention (tau$^2$ = 0; I$^2$ = 0%)* | | | |
| **Intervention class** | **SMD (95% CI)** | **Confidence in non-null effect** | **Ranking (p score)** |

(*Continued*)

**Table 5.** (Continued)

| AAP + SSRI | −0.57 (−1.34 to 0.20) | Very low | 1 (0.84) |
|---|---|---|---|
| SARI | −0.37 (−0.90 to 0.16) | Very low | 2 (0.73) |
| NMDA antagonist | −0.35 (−0.88 to 0.17) | Very low | 3 (0.73) |
| SSRI | −0.08 (−0.36 to 0.20) | Very low | 4 (0.46) |
| Glutamatergic antagonist | 0.02 (−0.84 to 0.80) | Very low | 5 (0.44) |
| Pharmacological placebo | — | — | 6 (0.37) |
| TeCA | 0.41 (−0.65 to 1.47) | Very low | 7 (0.22) |
| TCA | 0.48 (−0.78 to 1.72) | Very low | 8 (0.20) |

**Alcohol use at postintervention** ($tau^2 = 0.0149$; $I^2 = 20.5\%$)

| Intervention class | SMD (95% CI) | Confidence in non-null effect | Ranking (p score) |
|---|---|---|---|
| NMDA antagonist | −2.23 (−2.87 to −1.58) | Very low | 1 (0.999) |
| TeCA | −0.54 (−1.64 to 0.55) | Very low | 2 (0.62) |
| SSRI | −0.30 (−0.59 to −0.02) | Low | 3 (0.57) |
| SARI | −0.23 (−0.66 to 0.21) | Very low | 4 (0.48) |
| TCA | −0.09 (−0.66 to 0.49) | Very low | 5 (0.34) |
| Glutamatergic antagonist | −0.00 (−0.85 to 0.85) | Very low | 6 (0.29) |
| Pharmacological placebo | — | — | 7 (0.20) |

**Alcohol use at long-term follow-up** ($tau^2 = 0$; $I^2 = 0\%$)

| Intervention class | SMD (95% CI) | Confidence in non-null effect | Ranking (p score) |
|---|---|---|---|
| SSRI | −0.43 (−1.14 to 0.29) | Very low | 1 (N/A) |
| Pharmacological placebo | — | — | 2 (N/A) |

**Heavy drinking at postintervention** ($tau^2 = 0.0429$; $I^2 = 44\%$)

| Intervention class | SMD (95% CI) | Confidence in non-null effect | Ranking (p score) |
|---|---|---|---|
| SARI | −1.04 (−1.80 to −0.27) | Very low | 1 (0.94) |
| Opioid antagonist + SSRI | −0.60 (−1.13 to −0.08) | Very low | 2 (0.79) |
| NMDA antagonist | −0.58 (−1.25 to 0.09) | Very low | 3 (0.76) |
| SSRI | −0.13 (−0.43 to 0.17) | Very low | 4 (0.44) |
| Opioid antagonist | −0.09 (−0.50 to 0.33) | Very low | 5 (0.40) |
| Pharmacological placebo | — | — | 6 (0.30) |
| TCA | 0.22 (−0.45 to 0.88) | Very low | 7 (0.19) |
| TeCA | 0.39 (−0.74 to 1.52) | Very low | 8 (0.17) |

**Health-related quality of life at postintervention** ($tau^2 = 0$; $I^2 = 0\%$)

| Intervention class | SMD (95% CI) | Confidence in non-null effect | Ranking (p score) |
|---|---|---|---|
| SSRI vs. NMDA antagonist | −0.09 (−0.53 to 0.35) | Very low | N/A |
| Glutamatergic antagonist vs. placebo | −0.33 (−1.15 to 0.50) | Very low | N/A |

**Functional status at postintervention** ($tau^2 = 0$; $I^2 = 0\%$)

| Intervention class | SMD (95% CI) | Confidence in non-null effect | Ranking (p score) |
|---|---|---|---|
| NMDA antagonist | −1.21 (−1.84 to −0.59) | Very low | 1 (0.88) |
| AAP + SSRI | −1.02 (−1.86 to −0.18) | Very low | 2 (0.71) |
| SSRI | −0.92 (−1.36 to −0.47) | Moderate | 3 (0.64) |
| TCA | −0.83 (−1.36 to −0.29) | Very low | 4 (0.59) |
| SARI | −0.63 (−1.24 to −0.03) | Very low | 5 (0.45) |
| Glutamatergic antagonist | −0.07 (−0.89 to 0.75) | Very low | 6 (0.15) |
| Pharmacological placebo | — | — | 7 (0.08) |

**Functional status at long-term follow-up** ($tau^2 = 0$; $I^2 = 0\%$)

| Intervention class | SMD (95% CI) | Confidence in non-null effect | Ranking (p score) |
|---|---|---|---|
| SSRI | −0.70 (−1.42 to 0.03) | Very low | 1 (N/A) |
| Pharmacological placebo | — | — | 2 (N/A) |

(*Continued*)

**Table 5.** (Continued)

| *Adverse events at postintervention (tau² = 0.5338; I² = 54%)* | | | |
|---|---|---|---|
| **Intervention class** | **OR (95% CI)** | **Confidence in non-null effect** | **Ranking (*p* score)** |
| MRI | 0.43 (0.03 to 6.54) | Very low | 1 (0.78) |
| NMDA antagonist | 0.51 (0.03 to 8.26) | Very low | 2 (0.75) |
| nAChRs | 0.56 (0.05 to 5.76) | Very low | 3 (0.74) |
| Pharmacological placebo | — | — | 4 (0.67) |
| Opioid antagonist | 1.02 (0.11 to 9.26) | Very low | 5 (0.62) |
| SARI | 1.27 (0.36 to 4.49) | Very low | 6 (0.57) |
| Glutamatergic antagonist | 2.40 (0.26 to 21.96) | Very low | 7 (0.40) |
| SSRI | 2.20 (0.94 to 5.16) | Moderate | 8 (0.38) |
| TCA | 2.34 (0.64 to 8.61) | Very low | 9 (0.37) |
| TeCA | 3.46 (0.21 to 55.75) | Very low | 10 (0.33) |
| Opioid antagonist + SSRI | 4.80 (0.74 to 31.31) | Very low | 11 (0.21) |
| AAP + SSRI | 8.01 (0.44 to 145.26) | Very low | 12 (0.18) |
| *Adverse events at long-term follow-up (tau² = 0; I² = 0%)* | | | |
| **Intervention class** | **OR (95% CI)** | **Confidence in non-null effect** | **Ranking (*p* score)** |
| Pharmacological placebo | — | — | 1 (N/A) |
| SSRI | 1.06 (0.02 to 57.01) | Very low | 2 (N/A) |
| *Serious adverse events at postintervention (tau² = 0; I² = 0%)* | | | |
| **Intervention class** | **OR (95% CI)** | **Confidence in non-null effect** | **Ranking (*p* score)** |
| Opioid antagonist + SSRI | 0.35 (0.12 to 1.05) | Very low | 1 (0.88) |
| Opioid antagonist | 0.93 (0.39 to 2.20) | Very low | 2 (0.56) |
| Pharmacological placebo | — | — | 3 (0.53) |
| TCA | 1.00 (0.02 to 52.85) | Very low | 4 (0.51) |
| TeCA | 1.00 (0.02 to 57.31) | Very low | 5 (0.51) |
| SSRI | 1.55 (0.81 to 2.96) | Low | 6 (0.29) |
| NMDA antagonist | 3.18 (0.25 to 39.80) | Very low | 7 (0.22) |

AAP, atypical antipsychotic; CI, confidence interval; MRI, monoamine reuptake inhibitor; nAChR, nonselective noncompetitive antagonists of the nicotinic acetylcholine receptor; NMDA, N-methyl-D-aspartate; OR, odds ratio; SARI, serotonin antagonist and reuptake inhibitor; SMD, standardized mean difference; SSRI, selective serotonin reuptake inhibitor; TCA, tricyclic antidepressant; TeCA, tetracyclic antidepressant.

absent from the net heat plot. We are very uncertain about the existence (or not) of a non-null effect based on the available evidence (i.e., we have very low confidence in all effect estimates). However, after excluding studies with high risk of bias from the pharmacologic network, we identified an estimate of a small beneficial effect of SSRIs over placebos (OR = 0.69; 95% CI, 0.47 to 0.9998; 5 RCTs, $p = 0.049$), indicating that SSRIs may have increased remission from alcohol use (i.e., low confidence) had this been the default analysis. All other sensitivity analyses did not substantively differ from the primary analyses for remission from alcohol use.

**Depressive symptoms.** Twenty pharmacological intervention studies (71%) reported data on 11 intervention classes, while 6 psychological intervention studies (75%) reported data on 6 intervention classes. We detected significant heterogeneity in the pharmacologic network globally ($Q(12) = 20.82$, $p = 0.05$), which was due to significant within-design heterogeneity in comparisons of placebos versus SSRIs ($Q(6) = 15.38$, $p = 0.02$) rather than inconsistency between designs ($Q(2) = 3.53$, $p = 0.17$). Hotspots of inconsistency were absent from (and direct versus indirect effect estimates were in the same direction in) the net heat plot

**Table 6. Summary of findings table for the psychological intervention network.**

| *Remission from depression at postintervention (tau$^2$ = 0; I$^2$ = 0%)* | | | |
|---|---|---|---|
| **Intervention class** | **OR (95% CI)** | **Confidence in non-null effect** | **Ranking (*p* score)** |
| IPT vs. SP | 4.33 (0.39 to 48.61) | Very low | N/A |
| CBT vs. placebo | 0.47 (0.08 to 2.66) | Very low | N/A |
| *Remission from alcohol use at postintervention (tau$^2$ = 0; I$^2$ = 0%)* | | | |
| **Intervention class** | **OR (95% CI)** | **Confidence in non-null effect** | **Ranking (*p* score)** |
| SMS | 0.61 (0.24 to 1.55) | Very low | 1 (0.89) |
| Placebo | 1.53 (0.34 to 6.87) | Very low | 2 (0.43) |
| No additional treatment | — | — | 3 (0.18) |
| IPT vs. SP | 1.16 (0.29 to 4.69) | Very low | N/A |
| *Remission from alcohol use at short-term follow-up (tau$^2$ = 0; I$^2$ = 0%)* | | | |
| **Intervention class** | **OR (95% CI)** | **Confidence in non-null effect** | **Ranking (*p* score)** |
| SMS vs. placebo | 1.64 (0.55 to 4.91) | Very low | N/A |
| *Remission from alcohol use at long-term follow-up (tau$^2$ = 0; I$^2$ = 0%)* | | | |
| **Intervention class** | **OR (95% CI)** | **Confidence in non-null effect** | **Ranking (*p* score)** |
| Placebo | 0.47 (0.07 to 2.96) | Very low | 1 (0.81) |
| No additional treatment | — | — | 2 (0.35) |
| SMS | 1.00 (0.36 to 2.78) | Very low | 3 (0.33) |
| *Remission from alcohol use at very long-term follow-up (tau$^2$ = 0; I$^2$ = 0%)* | | | |
| **Intervention class** | **OR (95% CI)** | **Confidence in non-null effect** | **Ranking (*p* score)** |
| SMS vs. placebo | 3.55 (0.76 to 16.43) | Very low | N/A |
| *Depressive symptoms at postintervention (tau$^2$ = 0.1043; I$^2$ = 64.9%)* | | | |
| **Intervention class** | **SMD (95% CI)** | **Confidence in non-null effect** | **Ranking (*p* score)** |
| CBT | −0.84 (−1.05 to −0.63) | Moderate | 1 (0.87) |
| SMS | −0.21 (−0.67 to 0.25) | Very low | 2 (0.73) |
| No additional treatment | — | — | 3 (0.20) |
| Placebo | −0.65 (−1.48 to 0.18) | Very low | 4 (0.20) |
| IPT vs. SP | −0.35 (−1.13 to 0.42) | Very low | N/A |
| *Depressive symptoms at short-term follow-up (SMS: tau$^2$ = 0; I$^2$ = 0%) (CBT: tau$^2$ = 0.0229; I$^2$ = 35.3%)* | | | |
| **Intervention class** | **SMD (95% CI)** | **Confidence in non-null effect** | **Ranking (*p* score)** |
| SMS vs. placebo | −0.17 (−0.74 to 0.39) | Very low | N/A |
| CBT vs. no additional treatment | −0.88 (−2.93 to 1.18) | Very low | N/A |
| *Depressive symptoms at long-term follow-up (tau$^2$ = 0; I$^2$ = 0%)* | | | |
| **Intervention class** | **SMD (95% CI)** | **Confidence in non-null effect** | **Ranking (*p* score)** |
| CBT | −1.59 (−1.83 to −1.34) | Very low | 1 (1.00) |
| SMS | −0.19 (−0.66 to 0.29) | Very low | 2 (0.39) |
| No additional treatment | — | — | 3 (0.11) |
| *Alcohol use at postintervention (tau$^2$ = 0; I$^2$ = 0%)* | | | |
| **Intervention class** | **SMD (95% CI)** | **Confidence in non-null effect** | **Ranking (*p* score)** |
| CBT | −0.25 (−0.47 to −0.04) | Low | 1 (0.88) |
| SMS | −0.13 (−0.58 to 0.32) | Very low | 2 (0.66) |
| No additional treatment | — | — | 3 (0.38) |
| Placebo | 0.34 (−0.36 to 1.05) | Very low | 4 (0.09) |
| *Alcohol use at short-term follow-up (tau$^2$ = 0; I$^2$ = 0%)* | | | |
| **Intervention class** | **SMD (95% CI)** | **Confidence in non-null effect** | **Ranking (*p* score)** |
| SMS vs. placebo | −0.14 (−0.70 to 0.43) | Very low | N/A |
| CBT vs. no additional treatment | −0.10 (−0.31 to 0.12) | Very low | N/A |
| *Alcohol use at long-term follow-up (tau$^2$ = 0; I$^2$ = 0%)* | | | |

(*Continued*)

**Table 6.** (Continued)

| Intervention class | SMD (95% CI) | Confidence in non-null effect | Ranking (p score) |
|---|---|---|---|
| CBT | −0.18 (−0.39 to 0.04) | Very low | 1 (0.81) |
| SMS | −0.06 (−0.53 to 0.41) | Very low | 2 (0.47) |
| No additional treatment | — | — | 3 (0.23) |
| *Withdrawal and craving symptoms at postintervention (tau² = 0; I² = 0%)* | | | |
| **Intervention class** | **SMD (95% CI)** | **Confidence in non-null effect** | **Ranking (p score)** |
| Placebo | −0.24 (−0.95 to 0.47) | Very low | 1 (0.80) |
| No additional treatment | — | — | 2 (0.42) |
| SMS | 0.05 (−0.41 to 0.52) | Very low | 3 (0.28) |
| *Withdrawal and craving symptoms at short-term follow-up (tau² = 0; I² = 0%)* | | | |
| **Intervention class** | **SMD (95% CI)** | **Confidence in non-null effect** | **Ranking (p score)** |
| SMS vs. placebo | −0.46 (−1.03 to 0.12) | Very low | N/A |
| *Withdrawal and craving symptoms at long-term follow-up (tau² = 0; I² = 0%)* | | | |
| **Intervention class** | **SMD (95% CI)** | **Confidence in non-null effect** | **Ranking (p score)** |
| SMS vs. no additional treatment | −0.08 (0.58 to 0.42) | Very low | N/A |
| *Functional status at postintervention (tau² = 0; I² = 0%)* | | | |
| **Intervention class** | **SMD (95% CI)** | **Confidence in non-null effect** | **Ranking (p score)** |
| SMS vs. placebo | −0.97 (−1.54 to −0.41) | Very low | N/A |
| *Functional status at short-term follow-up (tau² = 0; I² = 0%)* | | | |
| **Intervention class** | **SMD (95% CI)** | **Confidence in non-null effect** | **Ranking (p score)** |
| SMS vs. placebo | −0.54 (−1.12 to 0.04) | Very low | N/A |

CBT, cognitive behavioral therapy; CI, confidence interval; IPT, interpersonal therapy; OR, odds ratio; SMS, self-management support.

investigating SSRIs versus placebo (OR from direct evidence = 0.11 and OR from indirect evidence = 0.49), TCAs versus placebo (OR from direct evidence = 0.52 and OR from indirect evidence = 0.11), and SSRIs versus TCAs (OR from direct evidence = 0.01 and OR from indirect evidence = 0.41). While did not detect significant heterogeneity in the psychological network globally ($Q(2) = 5.69$, $p = 0.0581$), we did detect significant inconsistency between designs ($Q(1) = 5.53$, $p = 0.0187$), although we were not able to assess hotspots of inconsistency in a net heat plot because of a lack of closed loops in the network geometry. Due to significant inconsistency between designs, we used meta-analytic estimates derived from only direct evidence (rather than network meta-analytic estimates based in part on indirect evidence) for psychological interventions with direct comparisons [43,45].

We have moderate confidence in an estimate at postintervention of at least a medium beneficial effect of CBTs over no treatment additional to TAU (SMD = −0.84; 95% CI, −1.05 to −0.63; 2 RCTs; $p < 0.001$); we downgraded confidence from "high" to "moderate" because the body of evidence only had 2 studies providing direct evidence for assessing consistency. We have low confidence in an estimate at postintervention of a non-null effect favoring TCAs over placebo (SMD = −0.37; 95% CI, −0.72 to −0.02; 3 RCTs; $p = 0.038$); we downgraded confidence from "high" to "low" because of a risk of attrition bias in the body of evidence, as well as concerns about intransitivity due to differences in length of treatment among studies in this network. We have very low confidence in all other estimates. In the ad hoc sensitivity analysis removing studies conducted prior to DSM-III, the CI for the estimate of TCAs versus placebo no longer excluded a null effect (SMD = −0.31; 95% CI, −0.71 to 0.10; 2 RCTs; $p = 0.137$). All other sensitivity analyses did not substantively differ from the primary analyses for depressive symptoms.

**Alcohol use.**   Nine pharmacological intervention studies (32%) reported data on 7 intervention classes, while 3 psychological intervention studies (38%) reported data on 4 intervention classes. We did not detect significant heterogeneity in the pharmacologic network globally ($Q(3) = 3.78$, $p = 0.29$), although we were not able to assess hotspots of inconsistency in a net heat plot because of a lack of closed loops in the network geometry. We have moderate confidence in an estimate at postintervention of a non-null effect favoring SSRIs over placebos (SMD = −0.30; 95% CI, −0.59 to −0.02; 3 RCTs; $p = 0.039$); we downgraded confidence from "high" to "moderate" because of a risk of selective outcome reporting. We have low confidence in an estimate at postintervention of a non-null effect favoring CBTs over no treatment additional to TAU (SMD = −0.25; 95% CI, −0.47 to −0.04; 1 RCT; $p = 0.021$); we downgraded confidence from "high" to "low" because the body of evidence only had one study providing direct evidence for assessing consistency. We have very low confidence in all other estimates. Sensitivity analyses did not substantively differ from the primary analyses for alcohol use.

**Heavy drinking.**   Nine pharmacological intervention studies (32%) reported data on 7 intervention classes, while no psychological intervention studies reported data on heavy drinking. We did not detect significant heterogeneity in the pharmacologic network globally ($Q(4) = 7.15$, $p = 0.13$); while there was significant within-design heterogeneity in comparisons of pharmacological placebos versus SSRIs ($Q(2) = 6.00$, $p = 0.05$), we did not detect inconsistency between designs ($Q(2) = 1.15$, $p = 0.56$). Hotspots of inconsistency were absent from the net heat plot. We have very low confidence in all effect estimates. Sensitivity analyses did not substantively differ from the primary analyses for heavy drinking.

**Withdrawal/craving symptoms.**   Eight pharmacological intervention studies (29%) reported data on 8 intervention classes, while 2 psychological intervention studies (25%) reported data on 3 intervention classes. We did not detect significant heterogeneity in the pharmacologic network globally ($Q(1) = 0.01$, $p = 0.91$), although we were not able to assess hotspots of inconsistency in a net heat plot because of a lack of closed loops in the network geometry. We have very low confidence in all estimates. Sensitivity analyses did not substantively differ from the primary analyses for withdrawal/craving symptoms.

**Health-related quality of life.**   Two pharmacological intervention studies (7%) reported data on 4 intervention classes, while no psychological intervention studies reported data on health-related quality of life. We have very low confidence in all estimates. Sensitivity analyses did not substantively differ from the primary analyses for health-related quality of life.

**Functional status.**   Nine pharmacological intervention studies (32%) reported data on 7 intervention classes, while 1 psychological intervention study (13%) reported data on 2 interventions. We did not detect significant heterogeneity in the pharmacologic network globally ($Q(3) = 1.76$, $p = 0.62$), although we were not able to assess hotspots of inconsistency in a net heat plot because of a lack of closed loops in the network geometry. We have moderate confidence in an estimate of at least a small beneficial effect at postintervention of SSRIs over placebos (SMD = −0.92; 95% CI, −1.36 to −0.47; 3 RCTs; $p < 0.001$); we downgraded confidence from "high" to "moderate" because of a risk of selective outcome reporting (i.e., "serious" limitations of included studies). We have very low confidence in all other estimates at postintervention. Sensitivity analyses did not substantively differ from the primary analyses for functional status.

**Adverse events.**   Seventeen pharmacological intervention studies (61%) reported adverse event data on 12 intervention classes, with 6 pharmacological intervention studies (2%) reporting serious adverse event data on 7 intervention classes. No psychological intervention studies reported adverse event data. We detected significant heterogeneity in the adverse event network globally ($Q(8) = 17.41$, $p = 0.03$), which was due to significant within-design heterogeneity in comparisons of placebos versus SARIs ($Q(1) = 11.55$, $p < 0.001$) rather than inconsistency between designs ($Q(1) = 0.66$, $p = 0.42$). We did not detect significant

heterogeneity in the serious adverse event network globally ($Q(2)$ = 0.62, $p$ = 0.74). We were not able to assess hotspots of inconsistency in a net heat plot for both adverse events and serious adverse events. We have moderate confidence that patients receiving SSRIs were more likely to experience an adverse event than patients receiving pharmacological placebos (OR = 2.20; 95% CI, 0.94 to 5.16; 6 RCTs; $p$ = 0.07). We downgraded confidence from "high" to "moderate" because of a wide CI that included the null effect (i.e., "serious" imprecision). However, we did not downgrade 2 levels (i.e., "very serious" imprecision), because the CI did not include the threshold for a meaningful reduction in the likelihood of experiencing an adverse event. In addition, we did not find sufficient reason to downgrade due to study limitations, indirectness, inconsistency, publication bias, intransitivity, or incoherence.

We also have low confidence that patients receiving SSRIs had a greater risk of experiencing a serious adverse event than patients receiving placebos (OR = 1.56; 95% CI, 0.81 to 2.94; 3 RCTs; $p$ = 0.184); we downgraded confidence from "high" to "low" because of a wide CI and a risk of attrition bias. We have very low confidence in all other estimates. However, after excluding studies with high risk of bias from the pharmacologic network, we identified an estimate excluding a null effect (OR 2.57; 95% CI, 1.30 to 5.08; 65 RCTs; $p$ = 0.007]), indicating patients receiving SSRIs were more likely to experience an adverse event than patients receiving placebos (i.e., high confidence) had this been the default analysis. All other sensitivity analyses did not substantively differ from the primary analyses for adverse events.

## Discussion

The available body of evidence on treatments for adults with both an alcohol use and depressive disorder includes 14 pharmacological interventions and 4 psychological interventions. These interventions represent a fraction of the interventions discussed and recommended in clinical practice guidelines for either alcohol use or depressive disorders [7,8]. Moreover, we have very low confidence in all estimates of intervention effects on our primary outcomes (i.e., remission from depression and remission from alcohol use). We also did not have high confidence in any effect estimates, and we have very low confidence in the vast majority of estimates of intervention effects across all outcomes. We are confident only in estimates at postintervention about the benefits of CBTs (on depressive symptoms and alcohol use), SSRIs (on functional status and alcohol use), and TCAs (on depressive symptoms) to be sufficient enough to warrant their consideration for policy and practice. Using language from the GRADE approach, CBTs likely reduced depressive symptoms (moderate confidence) and may have reduced alcohol use (low confidence), SSRIs likely improved functional status (moderate confidence) and may have reduced alcohol use (low confidence), and TCAs may have reduced depressive symptoms (low confidence). However, we also found SSRIs to have a higher risk of adverse events (including serious adverse events). Using language from the GRADE approach, patients receiving SSRIs likely had a greater risk of experiencing an adverse event compared with patients receiving pharmacological placebos (moderate confidence), and they may have had a greater risk of experiencing a serious adverse event (low confidence).

We have very low confidence in all other effect estimates (including for both of our primary outcomes and time points later than postintervention), meaning we are very uncertain about the existence (or not) of a non-null effect for all other outcomes, based on the available evidence. Our very low confidence in most effect estimates is primarily driven by sparse networks with limited data. While we identified almost 3 dozen trials, most trials were underpowered, almost all of the evidence on effects is at postintervention without longer-term follow-ups, and the networks of evidence for outcomes were sparse. Most bodies of evidence included only indirect evidence or direct evidence from only 1 or 2 studies. This absence of evidence on

interventions and very low confidence in effect estimates does not indicate evidence of an absence of effects, but rather that future studies are needed to overcome limitations in the current body of evidence (such as limited study duration and insufficient statistical power). Furthermore, given that identified effects in which we had at least low confidence were all at postintervention, applicability of evidence on drinking outcomes to inpatient and residential care settings may be limited.

The results of this review are comparable to the conclusions of previous reviews in this area. Previous reviews have found antidepressants to be more effective than placebo in treating depression among patients with comorbid AUDs [14,22], as well as finding clinical intervention in general (any form of medication or psychosocial treatment) for depression co-occurring with an AUD to be associated with an early improvement in depressive symptoms [20]. The most recent Cochrane review of antidepressants in the treatment of people with co-occurring depression and alcohol dependence found that antidepressants had positive effects on certain outcomes relevant to depression and drinking alcohol (e.g., remission from alcohol use and alcohol use) but not on other relevant outcomes (e.g., remission from depression and depressive symptoms), and the risk of developing adverse effects appeared to be minimal [49]. Moreover, a review on combined CBTs and motivational interviewing for patients with a depressive disorders and AUDs found small but clinically significant effects compared with TAU on depressive symptoms and alcohol consumption [23]. Our review builds on these previous studies through the use of NMA to provide estimates of the comparative effectiveness of specific intervention classes across a range of outcomes.

## Strengths and limitations

This review has several strengths: an a priori research design, duplicate study selection and data extraction of study information, a comprehensive search of electronic databases, and comprehensive assessments of confidence in the body of evidence used to formulate review conclusions. However, we did not contact trial authors for missing data or to find other potential studies not identified by the search strategy; additional outcome data (if existent), information about potential risks of bias, and other potential studies identified by trial authors have the potential to influence the effect estimates and confidence in the body of evidence. In addition, we used SMDs for estimating effects of continuous outcomes. While most data come from established measures for depressive symptoms, drinking, withdrawal and craving symptoms, quality of life, and functional status, the development and use of core outcome measurement sets for this clinical area would help allay concerns about the sensitivity of the direction and magnitude-of-effect estimates arising from application of suboptimal instruments [50]. In addition, several studies did not report important information about study methods needed to assess risk of bias as well as the study context (e.g., stage in clinical pathway and type of clinical setting) helpful to assessing applicability of findings. We also note that the definition of adverse events was heterogeneous across studies when reported; defining and analyzing adverse events in numerous can hinder the ability to compare the net benefit (i.e., the balance between desirable and undesirable health effects) [51] across interventions in systematic reviews [52]. Furthermore, as we did not identify any RCTs comparing a pharmacological to a psychological intervention, we had analyze these families of interventions in separate networks, thereby preventing us from drawing any comparisons between (classes of) pharmacological and psychological interventions. Consequently, we caution readers in any such comparisons they may make using the results of this review. Lastly, we conducted network meta-analyses using the class of intervention as the node; caution must be exercised in applying findings to individual interventions within a class, particularly for networks in which significant heterogeneity exists.

## Conclusions

Those charged with developing guidelines, providing recommendations for health systems, and treating patients may be interested in using these findings to inform policy and practice. We are very uncertain about the existence (or not) of any non-null effects for our primary outcomes of remission from depression and remission from alcohol use. We also did not have high confidence in any effect estimates, and we have very low confidence in the vast majority of estimates of intervention effects across all outcomes. The available evidence does suggest potentially actionable benefits for patients with both an AUD and a depressive disorder at postintervention of CBTs for depressive symptoms and alcohol use, TCAs for depressive symptoms, and SSRIs for alcohol use and functional status—although SSRIs also have higher risks of adverse events (including serious adverse events). However, these potentially actionable benefits only apply to postintervention and are not against active comparators, limiting understanding of the efficacy of interventions in the long term as well as the comparative effectiveness of active treatments. Future studies are needed to provide more conclusive evidence about the (comparative) effectiveness of clinical interventions for treating adults with depressive disorders and AUDs.

Researchers, policymakers, funders, and practitioners may wish to use findings to establish future priorities on researching clinical interventions for this patient population. In addition to seeking to replicate evidence underpinning the abovementioned potentially actionable benefits, future trials could prioritize direct comparisons of comparisons with effect estimates suggesting intervention superiority but for which we have insufficient confidence to support consideration for policy and practice recommendations on the basis of evidence on effectiveness. Examples include SSRIs on remission for alcohol use and depressive symptoms at long-term follow-up, and opioid antagonists in combination with SSRIs on remission for alcohol use, depressive symptoms, and heavy drinking at postintervention.

In addition to more studies on interventions included in this review, studies are needed on other interventions used to treat AUDs and depressive disorders. Examples of interventions missing from this body of evidence that are recommended in clinical practices guidelines for AUDs include 12-Step Facilitation, behavioral couples therapy, the community reinforcement approach, disulfiram, gabapentin, motivational enhancement therapy, and topiramate [7]. Examples of interventions missing from this body of evidence that are recommended in clinical practices guidelines for depressive disorders include 5-HT2 and 5-HT3 receptor antagonists, behavioral activation, monoamine oxidase inhibitors, mindfulness-based therapies, norepinephrine and dopamine reuptake inhibitors, problem-solving therapy, and serotonin and norepinephrine reuptake inhibitors [8].

To ensure their utility in overcoming limitations of the current body of evidence for informing policy and practice, researchers should design future studies that are adequately powered and fit for this pragmatic purpose [53], prospectively register fully developed protocols and statistical analysis plans [54,55], and comprehensively report completed trials [56,57]. Given concerns about use of some pharmacological interventions in patients with AUDs (due to potential interactions with medications and alcohol), this research area would also benefit from standards on the collection and reporting of adverse events [58].

## Disclaimers

The findings and conclusions in this manuscript are those of the authors and do not necessarily represent the views of the US Department of Defense Psychological Health Center of Excellence (https://www.pdhealth.mil/).

## Supporting information

**S1 Appendix. RMarkdown file with code for running network meta-analyses.**
(RMD)

**S2 Appendix. Excel file with study data.**
(XLSX)

**S1 PRISMA Checklist. Completed PRISMA-NMA checklist.** PRISMA-NMA, Preferred
Reporting Items for Systematic Reviews and Meta-Analyses Network Meta-Analyses.
(DOCX)

**S1 Text. Supporting information content containing the full study protocol and meta-analytic outputs.**
(DOCX)

## Author Contributions

**Conceptualization:** Sean Grant, Susanne Hempel.

**Data curation:** Sean Grant.

**Formal analysis:** Sean Grant, Marika Booth.

**Funding acquisition:** Susanne Hempel.

**Investigation:** Sean Grant, Gulrez Azhar, Eugeniu Han, Marika Booth, Jody Larkin.

**Methodology:** Sean Grant, Marika Booth, Aneesa Motala, Jody Larkin, Susanne Hempel.

**Project administration:** Sean Grant, Aneesa Motala, Susanne Hempel.

**Resources:** Sean Grant, Marika Booth, Jody Larkin, Susanne Hempel.

**Software:** Sean Grant, Marika Booth.

**Supervision:** Sean Grant.

**Validation:** Sean Grant, Marika Booth.

**Visualization:** Sean Grant.

**Writing – original draft:** Sean Grant, Gulrez Azhar, Eugeniu Han, Marika Booth, Aneesa
Motala, Jody Larkin, Susanne Hempel.

**Writing – review & editing:** Sean Grant, Gulrez Azhar, Eugeniu Han, Marika Booth, Aneesa
Motala, Jody Larkin, Susanne Hempel.

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
