## [Decision Letter · Decision Letter 0]

20 Feb 2020

Dear Dr. Grant,

Thank you very much for submitting your manuscript "Clinical interventions for adults with comorbid alcohol use and depressive disorders: A systematic review and network meta-analysis" (PMEDICINE-D-19-03407) for consideration at PLOS Medicine. 

[LINK]

In light of these reviews, I am afraid that we will not be able to accept the manuscript for publication in the journal in its current form, but we would like to consider a revised version that addresses the reviewers' and editors' comments. Obviously we cannot make any decision about publication until we have seen the revised manuscript and your response, and we plan to seek re-review by one or more of the reviewers. 

We expect to receive your revised manuscript by Mar 12 2020 11:59PM. Please email us (plosmedicine@plos.org) if you have any questions or concerns.

We look forward to receiving your revised manuscript. 

Sincerely,

Caitlin Moyer, Ph.D.

Associate Editor 

PLOS Medicine

plosmedicine.org

Comments from the Academic Editor:

1. R1 points out (comment #2) the limitations of aggregating findings across editions of the DSM. I am in agreement that this aggregation is problematic but feel less strongly that the solution is to exclude them. In my opinion, it would also be reasonable to conduct sensitivity analyses excluding the pre-DSM-III studies and reporting the findings of the sensitivity analyses in Appendix Tables.

2. Related to the above, the inclusion criteria on page 3 line 48 indicate that you included "parallel-group (individually- or cluster-) randomized controlled trials only. Studies had to include adult participants (at least 50% were 18 years of age or older) with clinical diagnoses for both an AUD and depressive disorder according to Diagnostic and Statistical Manual of Mental Disorders (DSM) or International Classification of Diseases (ICD) criteria." This sentence would suggest that inclusion was limited to samples where study participants had DSM-consistent diagnoses. However, Tables 1 & 2 seem to indicate that there were a number of (mostly pre-1980) studies included where ascertainment of AUD did not follow any version of the DSM (eg, Butterworth 1971, "alcoholic"; Zielinski 1979, "alcoholic", etc.). Can the authors please clarify? In the Appendix, the authors state that "In addition to formal diagnostic procedures, we also included studies that used non-operationalized diagnostic criteria, validated clinician-reported symptom questionnaires, or self-reported symptom questionnaires with established thresholds to identify patients with eligible diagnoses." This sentence would explain the inclusion of studies like Butterworth & Zielinski and should be included in the main text. It is also worth considering whether these pre-1980 studies that did not use structured diagnoses should also be excluded in sensitivity analyses (like the pre-DSM-III studies described above).

3. R2 highlights some sensitivity analyses that are unclearly reported. In my opinion, the findings of these sensitivity analyses can be reported in detail in Appendix Tables.

4. Please review the tables and appendix tables for errors. For example, Zielinski is identified as "Zielinski 1977" in the online supplement but "Zielinski 1979" in the main text.

5. Please include a list of the included studies in the Appendix. Throughout the main text and appendix materials, the studies are identified by "Author Year", but the studies themselves are not included in the list of eReferences. The list of included studies can be listed separately in the Appendix. Also relevant to R2's comment about 35 studies/86 citations, the Appendix of included studies can group together studies where findings are reported in multiple publications.

Editorial comments:

In the abstract please combine the methods and findings into one heading, per house style. Please also include p values where 95% Cis are used both in the abstract and throughout. Please include a sentence on the limitations of the study as the final sentence of the methods and findings section of the abstract.

Main text- ‘Background’ should be introduction.

Author Summary - At this stage, we ask that you include a short, non-technical Author Summary of your research to make findings accessible to a wide audience that includes both scientists and non-scientists. The Author Summary should immediately follow the Abstract in your revised manuscript. This text is subject to editorial change and should be distinct from the scientific abstract. Please see our author guidelines for more information: https://journals.plos.org/plosmedicine/s/revising-your-manuscript#loc-author-summary

Refs in the main text should be in square brackets rather than round and placed ahead of punctuation. 

Please ensure that the study is reported according to the PRISMA guideline, and include the completed PRISMA checklist as Supporting Information. When completing the checklist, please use section and paragraph numbers, rather than page numbers. Please add the following statement, or similar, to the Methods: "This study is reported as per the Strengthening the Reporting of Observational Studies in Epidemiology PRISMA guideline (S1 Checklist)."

Please report your study according to the relevant guideline, which can be found here: http://www.equator-network.org/

Comments from the reviewers:

Reviewer #1: This is a meta-analysis of clinical interventions for adults with comorbid alcohol use and depression disorder. The topic is important to the field since this comorbid combination is extremely common and empirically-informed treatment guidelines are needed. The meta-analytic approach used is meticulously described in a way that bespeaks the authors high technical expertise. The results are carefully contextualized in terms of confidence level, with penalties for the quality and consistency of the data upon which they are based. These are all important strengths. 

There are also some weaknesses and areas for further clarification/elaboration. 

1) We are told how confident we can be in each result in multiple prominent locations in the article. Easily grasped and intuitive terms such as "low, moderate and high confidence" allow the reader to easily understand the ratings. This stands in contrast to how effect sizes are described. It would be helpful if the effect sizes (ORs and SMDs) were also contextualized in the same easily grasped language; e.g., small, medium and large. If the work is to be of value to the field, the reader should be able to more easily understand the size of the effects using intuitive anchors (again, small, medium or large) and, where possible, variable or person specific references (e.g., number needed to treat; NNT). It's one thing to know how confident we should be in an effect and it's another thing to know if the size of the effect is large enough to affect policy/practice. Only by considering these parameters together can policy and clinical decisions be made in an optimally informed way.

2) Studies included in the analysis go back to 1971 when the DSM was only in its second edition. This is a potential weakness because the criteria for psychiatric diagnoses changed dramatically from DSM II to DSM III but then changed very little after DSM III. The reliability of pre-DSM III diagnoses were notoriously poor and served as a primary reason for the dramatic changes found in DSM III and beyond. Comparing DSM II diagnoses to diagnoses made by DSM III and later is comparing apples to oranges. Arguably, studies using diagnostic criteria that pre-date DSM III should not be included. 

3) Most effects that were identified with any confidence related to post-intervention rather than at follow up. This raises a question for studies that were conducted in residential AUD treatment; i.e., drinking is prohibited in residential care so how can drinking outcomes level or alcohol diagnoses at post treatment be considered meaningful in these studies?

4) 40% of studies involved outpatient care, 26% involved inpatient care and 11% involved both. This adds up to 78%. Similarly, 49% were conducted in treatment settings related to AUD, 3% in depression treatment settings and 26% in dual care; again, adding up to 78%. Earlier it says that only 71% reported this information (not 78%). Also, what about the 22% (or is it 29%) of the original studies that failed to state were their cases were ascertained? This would seem to be such a glaring omission in reporting, it could be argued that these studies should not be included. 

5) Most effects tested were reported with low confidence. We are warned that this absence of evidence is not the evidence of absence (a truism made famous by the infamous Donald Rumsfeld). The many low confidence findings should be understood, we are told, as indicating the need for future, presumably better studies; but this is inadequate. Arguably, one of the biggest implications of this work is that the 35 best studies of AUD-depression comorbidity treatment (i.e., those qualified to be included in the meta-analysis) are, at the end of the day, inadequate to yield confident answers to key questions. Perhaps one of the greatest services to the field these investigators could provide in this paper is to state clearly in the Discussion what past researchers have failed to do, and what future researchers need to do so the field can begin to build a database that yields confident answers to the central problems of the treatment of comorbidity. 

Reviewer #2: The purpose of this study was to evaluate the effectiveness of pharmacological and psychological interventions in patients with co-occurring AUDs and depressive disorders. The authors of the study conducted a systematic review and a network meta-analysis (NMA) of randomized controlled trials (RCTs). Overall, the methods are well described and consistent with the study protocol declared a priori on the PROSPERO database.

My main concern relates to the interpretation of the data and the conclusions of the study, which in my view do not accurately reflect the results of the study. The authors conclude that the available evidence suggests potential benefits of TCAs (tricyclic antidepressants) for depressive symptoms, and SSRIs (selective serotonin reuptake inhibitors) for total drinking and functional status. Although the authors rightly point out the need for additional studies to provide more conclusive evidence, my opinion is that this systematic review and NMA shows that there is no high-grade evidence for the use of pharmacological treatments in patients with co-occurring alcohol use disorders and depressive disorders. Regarding psychological interventions, the available data provided by the review are very limited, which prevents any conclusion on the comparative effectiveness of these interventions to be drawn. 

Main comments:

1) First, the confidence in effect sizes (assessed with the GRADE method) is very low for almost all comparisons (including psychological interventions), and the effect sizes observed on the remaining comparisons are inconsistent. For instance, the authors found a greater benefit of SSRIs over placebo for total drinking (SMD= -0.30, 95%CI -0.59 to -0.02), but not for the other outcomes related to alcohol consumption at post intervention (i.e. remission from alcohol use, craving symptoms, heavy drinking). This effect size was estimated from a network including only nine (32%) pharmacological intervention RCTs. What does the number of 3 RCTs mentioned with the effect size correspond to (p10 l208)? Does is correspond to the number of studies providing a direct comparison between SSRIs and placebo? In that case, it would represent about one third of the studies evaluating SSRI vs placebo. However, there was no evidence of a superiority of SSRIs at long term follow-up (Table 4), and no evidence of a superiority of SSRIs over other active treatments (eTable 5, 6 and 7). The same concerns can be raised for the tricyclic antidepressants for which the superiority over placebo on depressive symptoms (SMD= -0.37, 95%CI -0.72 to -0.02; low confidence) was not observed on other outcomes, whereas there was no evidence of a superiority of TCAs over other active treatments.

Second, about half of the studies had an unclear or high risk of bias related to completeness of outcome data at post intervention. As a consequence, an attrition bias cannot be ruled out. Further to this, the majority of studies included in the review had a length of treatment ≤ 12 weeks, restricting the interpretation of the results over the long term.

Third, the potential benefit of a treatment cannot be considered without discussing its safety profile. The authors rightly highlighted the higher risk of adverse events with SSRIs (OR=2.20, 95%CI 0.94 to 5.16). The use of antidepressants in patients with alcohol use disorders is a cause for concern, as mixing antidepressants and alcohol can lead to significant side effects (such as dizziness, drowsiness). According to table 3, the definition of the outcome related to adverse events is heterogeneous from one study to another (e.g. proportion who dropped out because of adverse events, mean SAFTEE score at post intervention, proportion with a treatment emergent adverse event, proportion who experienced transient drowsiness) which may have introduced bias in the evaluation of this criterion. It is also likely that the absence of difference

between the other active treatments and placebo could be explained by insufficient study duration, insufficient power and by poor quality in the reporting of harmful effects or events.

Finally, the results observed for a class of drugs taken as a whole cannot be extrapolated to each of the drug of this therapeutic class, and especially in case of heterogeneity in the network.

All of these limitations should be carefully discussed in the manuscript to avoid misinterpretation of the results by the reader.

Minor points:

2) In the methods, the authors stated that they "conducted sensitivity analyses that excluded pharmacological interventions that do not have legal approval to be prescribed in the United States, use alternative outcome data reported in included studies, and are based on risk of bias assessments". Could the authors reword the sentence and clarify how these analyses were conducted? However, the results of these analyses are not described, except for the sensitivity analysis regarding remission from alcohol use after excluding studies with high risk from the pharmacologic network (p8 l184: please change 0.9998 by 1.00).

3) According to figure 1 (flow diagram), 86 citations were included, corresponding to 35 studies. Could the authors clarify the difference between the number of citations and the number of studies included? The number of citations identified through database searching should be provided before removing duplicates. The number of RCTs evaluating pharmacological interventions and psychological interventions should also be given.

4) The table summarizing risk of bias across studies (eTable 4) should be included in the body of the manuscript instead of the appendix. Some of the fields are marked with an asterisk but the legend is missing.

5) In tables 4 and 5, please provide the I2 of the network for each outcome. In table 4, please remove the column "Interpretation of findings", which is misleading.

6) In figure 2 (network geometry), please provide a legend explaining the meaning of the width of the edges as well as the shaded area.

6) For clarity, the citations corresponding to each RCT included in the review should be referenced.

Reviewer #3: I confine my remarks to statistical aspects of this paper. These were fine, but I would like to see more detail about pairwise and network meta-analysis. These are fairly esoteric methods and will be unfavmilar to most readers.

Peter Flom

[LINK]

---

## [Decision Letter · Decision Letter 1]

24 Nov 2020

Dear Dr. Grant,

Thank you very much for submitting your revised manuscript "Clinical interventions for adults with comorbid alcohol use and depressive disorders: A systematic review and network meta-analysis" (PMEDICINE-D-19-03407R1) for consideration at PLOS Medicine. We do apologize for the long delay in sending you a response. 

Your paper was discussed with our academic editor, and sent to two of the previous reviewers, including a statistical reviewer. The reviews are appended at the bottom of this email and any accompanying reviewer attachments can be seen via the link below:

[LINK]

In light of these reviews, we will not be able to accept the manuscript for publication in the journal in its current form, but we would like to invite you to submit a revised version that addresses the reviewers' and editors' comments fully. You will appreciate that we cannot make a decision about publication until we have seen the revised manuscript and your response, and we expect to seek re-review by one or more of the reviewers. 

We hope to receive your revised manuscript by Dec 15 2020 11:59PM. Please email us (plosmedicine@plos.org) if you have any questions or concerns.

Please let me know if you have any questions. Otherwise, we look forward to receiving your revised manuscript in due course. 

Sincerely,

Richard Turner PhD

rturner@plos.org

Please update the search to a point in the past 3 months, say.

Please add a sentence to your abstract to summarize where the constituent studies were done and the timeframe, and the median study size. 

In the abstract and elsewhere, you discuss adverse events in patients receiving SSRIs, with a p=0.07. In that this finding does not appear to be statistically significant, please explain why "moderate confidence" is ascribed to both this finding and the apparent benefit of SSRIs for functional status (p<0.001). Is the word "significantly" at line 331 appropriate?

In the final sentence of the "Methods and Findings" subsection of your abstract, please limit this to 2-3 study limitations. 

Please use "White" rather than "Caucasian", e.g., at line 199.

Do you see any issues in comparing data from RCTs involving drugs, often placebo controlled, with those from RCTs involving cognitive therapies, where there is often no control for attention? We might have missed this in the current paper, but if not you may wish to address this in the discussion section. 

Please do not use italics for emphasis.

Throughout the paper, please remove spaces from within the square brackets for reference call-outs (e.g., "... depressive disorder [6,14].".

Please revisit "Author" in references 4 & 5, which should perhaps be removed. 

Please adapt the header for figure 1 to "Study flow diagram" or similar. 

Comments from the reviewers:

*** Reviewer #2: 

The authors have done a good job responding to my comments and comments from other reviewers. However, I remain concerned about the presentation of the results, particularly in the abstract.

The results of the primary outcomes (remission from depression and remission from alcohol use) should explicitly be described in the abstract, in order to avoid any misinterpretation of the results and 'spin'.

The fact that this review found no high grade evidence for the use of pharmacological and psychological interventions in patients with co-occurring AUDs and depressive disorders should also be highlighted. In my view, it is the key message to policy makers that should be put forward.

Some of the limitations of the study do not need to be highlighted in the abstract, such as the lack of contact with some trial authors for missing information and data, or the use of SMDs rather than mean differences. Conversely, crucial limitations preventing a clear interpretation of the data are missing, namely the absence of demonstrated treatment effect over the long-term, and an attrition bias that cannot be ruled out (half of the studies had an unclear or high risk of bias related to completeness of outcome data at post intervention).

*** Reviewer #3: 

I confine my remarks to statistical aspects of this paper. In general, these were very thorough and well done and I have only some minor issues to resolve before i can recommend publication.

The biggest one is defining networks. Maybe I missed it (but I did search the document for the term "NetworK" and I did look at S1) but I couldn't figure out how studies were assigned to networks or what they were for.

In addition, I'd prefer a little less emphasis on tests of significance in the heterogeneity analysis. I like the use of heat maps and I think the authors should emphasize the effect size -- how different were the studies? -- rather than whether the differences were significant.

But, overall, a very good job.

Peter Flom

***

[LINK]

---

## [Decision Letter · Decision Letter 2]

15 Sep 2021

Dear Dr. Grant,

Thank you very much for re-submitting your manuscript "Clinical interventions for adults with comorbid alcohol use and depressive disorders: A systematic review and network meta-analysis" (PMEDICINE-D-19-03407R2) for consideration at PLOS Medicine. We do apologize for the long delay in sending you a decision. 

I have discussed the paper with editorial colleagues and our academic editor, and it was also seen again by one reviewer. I am pleased to tell you that, once the remaining editorial and production issues are fully dealt with, we expect to be able to accept the paper for publication in the journal.

[LINK]

Please let me know if you have any questions, and we look forward to receiving the revised manuscript.   

Sincerely,

Richard Turner Ph.D.

rturner@plos.org

Requests from Editors:

Please use the form "non-null" throughout.

Please abbreviate journal names in your reference list, including "PLoS ONE" and "PLoS Med.".

Please re-label the attached checklist "S1_PRISMA_Checklist", and use this label in the text. 

Please confirm that no display items have been used in previous publications (in order to be published under a CC BY licence, they must be free of copyright). 

Comments from Reviewers:

*** Reviewer #3: 

The authors have addressed my concerns and I now recommend publication

Peter Flom

***

[LINK]

---

## [Editor Report · Decision Letter 3]

22 Sep 2021

Dear Dr Grant, 

On behalf of my colleagues and the Academic Editor, Dr Tsai, I am pleased to inform you that we have agreed to publish your manuscript "Clinical interventions for adults with comorbid alcohol use and depressive disorders: A systematic review and network meta-analysis" (PMEDICINE-D-19-03407R3) in PLOS Medicine.

Prior to final acceptance, please break the long summary point into two (aiming for three subsections, each of three to four short points, in the summary points as a whole). 

PRESS

Sincerely, 

Richard Turner, PhD 

rturner@plos.org